# Simultaneous Online System Identification and Control using Composite Adaptive Lyapunov-Based Deep Neural Networks

## Abstract

Although deep neural network (DNN)-based controllers are popularly used to control uncertain nonlinear dynamic systems, most results use DNNs that are pre-trained offline and the corresponding controller is implemented post-training. Recent advancements in adaptive control have developed controllers with Lyapunov-based update laws (i.e., control and update laws derived from a Lyapunov-based stability analysis) for updating the DNN weights online to ensure the system states track a desired trajectory. However, the update laws are based on the tracking error, and offer guarantees on only the tracking error convergence, without providing any guarantees on system identification. This paper provides the first result on simultaneous online system identification and trajectory tracking control of nonlinear systems using adaptive updates for all layers of the DNN. A combined Lyapunov-based stability analysis is provided, which guarantees that the tracking error, state-derivative estimation error, and DNN weight estimation errors are uniformly ultimately bounded. Under the persistence of excitation (PE) condition, the tracking and weight estimation errors are shown to exponentially converge to a neighborhood of the origin, where the rate of convergence and the size of this neighborhood depends on the gains and a factor quantifying PE, thus achieving system identification and enhanced trajectory tracking performance. As an outcome of the system identification, the DNN model can be propagated forward to predict and compensate for the uncertainty in dynamics under intermittent loss of state feedback. Comparative simulation results are provided on a two-link manipulator system and an unmanned underwater vehicle system with intermittent loss of state feedback, where the developed method yields significant performance improvement compared to baseline methods.

## 1 Introduction

Deep neural network (DNN)-based methods have garnered popularity as a means for identification and control of uncertain nonlinear dynamic systems. Traditional DNN-based control techniques involve initial offline system identification based on datasets gathered from experimental trials (Abbeel, Coates, and Ng, 2010; Bansal, Akametalu, Jiang, Laine, and Tomlin; Karg and Lucia, 2020; Li, Qian, Zhu, Bao, Helwa, and Schoellig, 2017; Punjani and Abbeel, 2015; Shi, Shi, OConnell, Yu, Azizzadenesheli, Anandkumar, Yue, and Chung, 2019; Zhou, Helwa, and Schoellig, 2017). Subsequently, the identified DNN is used as a model to design controllers using traditional model-based control techniques. However, the weight estimates of the DNN are fixed and not updated during task execution, raising questions about the model's reliability and adaptability. Moreover, it is often implicitly assumed that minimizing a loss function would result in the DNN identifying the system dynamics. Whether a system model can be identified depends on whether the system trajectories generate information sufficient for the model to be identified, which manifests in terms of the persistence of excitation (PE) condition on the model (Sastry and Bodson, 1989). Although the PE condition is well-studied and understood in the system identification literature for linear regression models, only a few recent works remark on the PE condition for DNNs (Lamperski, 2022; Nar and Sastry, 2019;2; Sridhar, Sokolsky, Lee, and Weimer, 2022). If the model is not identified, the DNN may not generalize its performance well beyond the explored trajectories. Consequently,

the controller may not accurately compensate for the uncertainty, thus hazarding the stability of the closed-loop system.

Recent results in (Griffis, Patil, Bell, and Dixon, 2023; Hart, Griffis, Patil, and Dixon, 2024; Joshi, Virdi, and Chowdhary, 2020; Le, Greene, Makumi, and Dixon, 2022a; Le, Patil, Nino, and Dixon, 2022b; Muthirayan and Khargonekar, 2023; Patil, Le, Greene, and Dixon, 2022a; Sun, Greene, Le, Bell, Chowdhary, and Dixon, 2022) offer online weight updates for the DNN-based controllers to achieve tracking error convergence. The online weight update laws are derived from a Lyapunov-based stability analysis, and the corresponding controllers are popularly known as Lyapunov-based (Lb)-DNN controllers. These results can achieve tracking error convergence regardless of whether the PE condition is satisfied. However, the update laws in these results are based only on tracking error feedback and are primarily meant to achieve tracking error convergence. Since the parameter update law converges to zero upon convergence of the tracking error in such results, it is difficult to draw conclusions regarding the accuracy of the parameter estimate. To address this problem, incorporating a prediction error, i.e., a measure of the discrepancy between the actual dynamics and their DNN-based estimate, into the adaptation law can help with parameter estimation. It is desirable to estimate the DNN parameters to achieve system identification in addition to trajectory tracking, where the identified model can be used to perform new tasks. For example, the identified model can be used to predict and compensate for the uncertain dynamics under intermittent loss of feedback (Bell, Sun, Volle, Ganesh, Nivison, and Dixon, 2023; Chen, Bell, Deptula, and Dixon, 2019; Pulido, Volle, Waters, Bell, Ganesh, and Shin, 2024). However, the prediction error is difficult or often impossible to obtain since the dynamics are unknown and the state-derivative is typically either unavailable or noisy.

The classical result in (Slotine and Li, 1989) develops adaptive controllers with a composite adaptation law that incorporates both tracking and prediction errors for nonlinear systems with linear-in-parameters (LIP) uncertainties, where a low-pass filter is applied on both sides of the dynamics to eliminate the unknown state-derivative term. However, extending the composite adaptation law from (Slotine & Li, 1989) to nonlinear-in-parameters (NIP) uncertainties such as DNNs is challenging because the inner-layer weights are embedded in nonlinear activation functions in a nested fashion. Thus, when a low-pass filter is applied to the dynamics, the resultant expression is not separable in terms of the model parameters, which introduces technical challenges as detailed in Appendix B.2.

**Main Contributions.** This paper provides the first result on simultaneous online system identification and trajectory tracking control of nonlinear systems using online updates for all layers of the DNN. The development involves a composite adaptation law based on a new prediction error formulation using a dynamic state-derivative observer, which is combined with the tracking error to construct a least squares-based composite adaptation law. To address the challenges posed by the nested and NIP structure of DNNs, the Jacobian of the DNN is used in a composite adaptation law. Then, a first-order Taylor series expansion of the DNN is used in the analysis to express the prediction error in terms of the parameter estimation error. Since the adaptation laws are tightly coupled with the observer and system dynamics, a combined Lyapunov-based stability analysis is performed which guarantees the tracking, observer, and parameter estimation errors are uniformly ultimately bounded (UUB). If the PE condition is satisfied, the tracking and weight estimation errors are shown to exponentially converge to a neighborhood of the origin. Specifically, guarantees on estimating the ideal DNN parameters imply accurate system identification. Thus, the identified DNN model can generalize beyond the points encountered by the system trajectory. As a result, the composite adaptive model is suitable for systems involving intermittent loss of state feedback, where the identified DNN model can be propagated forward in time to predict the uncertain dynamics when feedback is lost, under developed sufficient dwell-time conditions. To demonstrate the performance and efficacy of the developed method on different systems, comparative simulation results are provided on two systems: a robot manipulator and an unmanned underwater vehicle (UUV) with intermittent loss of state feedback. The developed composite adaptive Lb-DNN controller yields significant performance improvement when compared to the tracking error-based adaptive Lb-DNN in (Patil et al., 2022a) and state-derivative observer-based disturbance rejection controllers as baseline methods.

## 2    PROBLEM FORMULATION AND CONTROL DESIGN

Consider the second order nonlinear system

$$\ddot{x} \quad = \quad f(x,\dot{x}) + g(x,\dot{x})u, \tag{1}$$

where $x, \dot{x} \in \mathbb{R}^n$ denote the states with available measurements, $\ddot{x} \in \mathbb{R}^n$ is the unknown state-derivative, $f : \mathbb{R}^n \times \mathbb{R}^n \to \mathbb{R}^n$ denotes an unknown continuously differentiable drift function, $g : \mathbb{R}^n \times \mathbb{R}^n \to \mathbb{R}^{n \times m}$ denotes a known locally Lipschitz control effectiveness matrix, and $u \in \mathbb{R}^m$ denotes the control input. Let the tracking error $e \in \mathbb{R}^n$ be defined as

$$e \quad \triangleq \quad x - x_d(t), \tag{2}$$

where $x_d : \mathbb{R}_{\geq 0} \to \mathbb{R}^n$ denotes a smooth reference trajectory that is designed to satisfy $\|x_d(t)\| \leq \overline{x_d}$, $\|\dot{x}_d(t)\| \leq \overline{\dot{x}_d}$, and $\|\ddot{x}_d(t)\| \leq \overline{\ddot{x}_d}$ where $\overline{x_d}, \overline{\dot{x}_d}, \overline{\ddot{x}_d} \in \mathbb{R}_{>0}$ are user-selected constants. The control objective is to design a DNN-based adaptive controller with a composite adaptation law that yields UUB tracking and parameter estimation errors. To aid the subsequent development, the following assumption is made.

**Assumption 1.** *The function $g$ is full row rank, and its right pseudoinverse $g^+ : \mathbb{R}^n \times \mathbb{R}^n \to \mathbb{R}^{m \times n}$ given by $g^+(x,\dot{x}) \triangleq g(x,\dot{x})^\top \left(g(x,\dot{x})g(x,\dot{x})^\top\right)^{-1}$ is assumed to be bounded.*

Assumption 1 implies the system is not under-actuated. Many electromechanical systems satisfy this assumption, e.g., the robot manipulator and UUV considered in Section 4 of this paper, Stewart platforms, hexapod robots, etc. The developed method can be extended on a case-by-case basis to underactuated systems using standard nonlinear control tools (e.g., backstepping) unique for such underactuated systems. Because a universal closed-form stabilizing nonlinear controller cannot obtained for an arbitrary underactuated system even with perfect model knowledge, the derivation has to be done on a case-by-case basis for each specific underactuated system, depending on how $g$ is structured. For more information on performing such an extension, the extension to nonholonomic mobile robots is provided in Appendix F.

### 2.1    CONTROL DEVELOPMENT

To facilitate the control development, let the auxiliary error $r \in \mathbb{R}^n$ be defined as

$$r \quad \triangleq \quad \dot{e} + \alpha_1 e, \tag{3}$$

where $\alpha_1 \in \mathbb{R}_{>0}$ denotes a constant control gain. Taking the time-derivative on both sides of (3), and substituting (1)-(3) yields

$$\dot{r} \quad = \quad f(x,\dot{x}) + g(x,\dot{x})u - \ddot{x}_d(t) + \alpha_1 \left(r - \alpha_1 e\right). \tag{4}$$

DNNs are a powerful tool for approximating unstructured uncertainties, such as $f$, based on their universal function approximation capabilities (Kidger and Lyons, 2020). Consider the compact set $\Omega \triangleq \{\zeta \in \mathbb{R}^{2n} : \|\zeta\| \leq (\alpha_1 + 2)\chi + \overline{x_d} + \overline{\dot{x}_d}\}$, where $\chi \in \mathbb{R}_{>0}$ is a user-selected constant that defines bounds on signals defined in the subsequent development. Additionally, let $\Phi : \mathbb{R}^{2n} \times \mathbb{R}^p \to \mathbb{R}^n$ denote a general DNN architecture, where $p \in \mathbb{Z}_{>0}$ denotes the total number of DNN parameters. According to the universal function approximation theorem, the function space of DNNs is dense in $\mathcal{C}(\Omega)$, where $\mathcal{C}(\Omega)$ denotes the space of functions continuous over $\Omega$ (Kidger & Lyons, 2020). Thus, given a prescribed accuracy $\overline{\varepsilon} \in \mathbb{R}_{>0}$, there exists a DNN $\Phi$ with ideal weights $\theta^* \in \mathbb{R}^p$ such that $\sup_{X \in \Omega} \|f(x,\dot{x}) - \Phi(X,\theta^*)\| \leq \overline{\varepsilon}$, where $X \triangleq \begin{bmatrix} x^\top & \dot{x}^\top \end{bmatrix}^\top$. Therefore, the drift function can be modeled as

$$f(x,\dot{x}) \quad = \quad \Phi(X,\theta^*) + \varepsilon(X), \tag{5}$$

where $\varepsilon : \mathbb{R}^{2n} \to \mathbb{R}^n$ denotes an unknown function reconstruction error that can be bounded as $\sup_{X \in \Omega} \|\varepsilon(X)\| \leq \overline{\varepsilon}$. The following typical assumption is made to aid the subsequent development (cf., (Lewis, Yegildirek, and Liu, 1996a, Assumption 1)).

**Assumption 2.** *There exists a known constant $\overline{\theta} \in \mathbb{R}_{>0}$ such that the unknown ideal weights can be bounded as $\|\theta^*\| \leq \overline{\theta}$.*

Assumption 2 is reasonable because, in practice, the user can select $\bar{\theta}$ *a priori* to restrict the parameter search space. If such a selection does not obey Assumption 2, the selection may no longer allow the user to make $\bar{\varepsilon}$ arbitrarily small as guaranteed by the universal function approximation property. However, a bound $\bar{\varepsilon}$ satisfying $\sup_{X \in \Omega} \|\varepsilon(X)\| \leq \bar{\varepsilon}$ still exists due to the continuity of $f$ and $\Phi$ over $\Omega$. Using heuristic approaches, if such $\bar{\varepsilon}$ is found to be larger than the maximum allowable error, then $\bar{\theta}$ can be iteratively increased until it achieves the prescribed $\bar{\varepsilon}$. Notably, DNN architectures that contain spectral normalization layers (e.g., in (Shi et al., 2019)) inherently involve bounded ideal weights since the weight matrices are normalized by their spectral norms.

Based on (4) and the subsequent analysis, the control input is designed as

$$u = g^+(x, \dot{x}) \left( \ddot{x}_d(t) - (\alpha_1 + k_r) r + (\alpha_1^2 - 1) e - \Phi\left(X, \hat{\theta}\right) \right), \tag{6}$$

where $k_r \in \mathbb{R}_{>0}$ denotes a constant control gain, and $\hat{\theta} \in \mathbb{R}^p$ denotes the adaptive estimate of the DNN weights $\theta^*$ that is developed using subsequently designed adaptation laws. Substituting (5) and (6) into (4) yields

$$\dot{r} = \Phi(X, \theta^*) - \Phi\left(X, \hat{\theta}\right) + \varepsilon(X) - e - k_r r. \tag{7}$$

## 2.2 COMPOSITE ADAPTATION LAW

The classical result in (Slotine & Li, 1989) develops a composite adaptation law using tracking and prediction errors for robot manipulators that involve linearly parameterized uncertainties in the absence of exogenous disturbances. However, for NIP uncertainties such as DNNs, the traditional development of the prediction error is not applicable and a new approach is required. Hence, an innovation of this paper is a new prediction error formulation based on a dynamic state-derivative estimator that provides an estimate of the ground truth value of the drift $f$ (c.f., (Kamalapurkar, Reish, Chowdhary, and Dixon, 2017)). The dynamic state-derivative observer is designed as

$$\dot{\hat{r}} = g(x, \dot{x})u - \ddot{x}_d(t) + \alpha_1 (r - \alpha_1 e) + \hat{f} + \alpha_2 \tilde{r}, \qquad \dot{\hat{f}} = k_f \left( \dot{\hat{r}} + \alpha_2 \tilde{r} \right) + \tilde{r}, \tag{8}$$

where $\hat{r}, \hat{f} \in \mathbb{R}^n$ denote the observer estimates of $r$ and $f$, respectively, $\tilde{r}, \tilde{f} \in \mathbb{R}^n$ denote the observer errors $\tilde{r} \triangleq r - \hat{r}$ and $\tilde{f} \triangleq f(x, \dot{x}) - \hat{f}$, respectively, and $\alpha_2, k_f \in \mathbb{R}_{>0}$ denote constant observer gains. As is typical of observer designs, observer error $\tilde{r}$ is known because $r$ and $\hat{r}$ are known, and is used as feedback to the observer in (9) to estimate $f$. Since $\dot{\hat{r}}$ is unknown, (8) can be implemented by integrating both sides and using the relation $\int_{t_0}^t \dot{\hat{r}}(\tau)\,d\tau = \tilde{r}(t) - \tilde{r}(t_0)$ to obtain $\hat{f}(t) = \hat{f}(t_0) + k_f \tilde{r}(t) - k_f \tilde{r}(t_0) + \int_{t_0}^t (k_f \alpha_2 + 1)\tilde{r}(\tau)d\tau$, where $t_0$ denotes the initial time. Note that although $\hat{f}$ generated by the state-derivative estimator can also be used to compensate for $f$, such an approach results in a robust high-gain design which does not achieve the system identification objective and can cause large overshoots in the control input.

Taking the time-derivative of $\tilde{r}$ and $\tilde{f}$ and substituting their definitions along with (4) and (8) yields

$$\dot{\tilde{r}} = \tilde{f} - \alpha_2 \tilde{r}, \qquad \dot{\tilde{f}} = \dot{f} - k_f \tilde{f} - \tilde{r}, \tag{9}$$

where $\dot{\tilde{f}}$ is derived after substituting in $\dot{\tilde{r}}$. Using the dynamic state-derivative estimator, the prediction error $E \in \mathbb{R}^n$ is designed as

$$E \triangleq \hat{f} - \Phi\left(X, \hat{\theta}\right). \tag{10}$$

Then, the composite least squares adaptation law is designed as

$$\dot{\hat{\theta}} = \mathrm{proj}\left( -k_{\hat{\theta}} \Gamma(t)\hat{\theta} + \Gamma(t)\Phi'^{\top}\left(X, \hat{\theta}\right)(r + \alpha_3 E) \right), \tag{11}$$

where $\mathrm{proj}(\cdot)$ denotes a continuous projection operator (cf. (Krstic, Kanellakopoulos, and Kokotovic, 1995, Appendix E)) which ensures $\hat{\theta}(t) \in \mathcal{B}_{\bar{\theta}} \triangleq \{\theta \in \mathbb{R}^p : \|\theta\| \leq \bar{\theta}\}$ for all $t \in \mathbb{R}_{\geq 0}$, $\alpha_3, k_{\hat{\theta}} \in \mathbb{R}_{>0}$ denote constant gains, $\Phi'\left(X, \hat{\theta}\right) \in \mathbb{R}^{n \times p}$ denotes the Jacobian $\Phi'\left(X, \hat{\theta}\right) \triangleq \frac{\partial \Phi(X, \hat{\theta})}{\partial \hat{\theta}}$. The reader is referred to Appendix C for details on calculation of the Jacobian for a fully-connected

DNN. Similar development could also be used to derive the Jacobian for other DNN architectures. The term $\Gamma: \mathbb{R}_{\geq 0} \to \mathbb{R}^{p \times p}$ denotes a positive-definite (PD) time-varying least squares adaptation gain matrix that is a solution to (Slotine & Li, 1989, Eqns. (16) and (17))

$$\frac{d}{dt}\Gamma^{-1}(t) = -\beta(t)\Gamma^{-1}(t) + \Phi'^{\top}\left(X, \hat{\theta}\right)\Phi'\left(X, \hat{\theta}\right), \tag{12}$$

with the bounded-gain time-varying forgetting factor $\beta : \mathbb{R}_{\geq 0} \to \mathbb{R}_{\geq 0}$ designed as

$$\beta(t) \triangleq \beta_0 \left(1 - \frac{\|\Gamma(t)\|}{\varkappa_0}\right), \tag{13}$$

where $\beta_0, \varkappa_0 \in \mathbb{R}_{>0}$ are user-defined constants denoting the maximum forgetting rate and the bound on $\lambda_{\max}\{\Gamma(t)\}$, respectively. The adaptation gain in (12) is initialized to be PD such that $\|\Gamma(t_0)\| < \varkappa_0$, and it can be shown that $\Gamma(t)$ remains PD for all $t \in \mathbb{R}_{\geq t_0}$ (Slotine & Li, 1989). Since $\Gamma(t)$ is PD, there exists a constant $\varkappa_1 \in \mathbb{R}_{>0}$ such that $\lambda_{\min}\{\Gamma(t)\} \geq \varkappa_1$ for all $t \in \mathbb{R}_{\geq t_0}$. The term $\beta(t)$ can be lower bounded as $\beta \geq \beta_1$, where $\beta_1 \in \mathbb{R}_{\geq 0}$ is a constant which satisfies the properties stated in the following remark.

*Remark* 1. Consider the case when $\Phi'\left(X, \hat{\theta}\right)$ satisfies the PE condition, i.e., there exists constants $\varphi_1, \varphi_2 \in \mathbb{R}_{>0}$ for all $\underline{t} \in \mathbb{R}_{\geq t_0}$ and some $T \in \mathbb{R}_{>0}$ such that $\varphi_1 I_p \leq \int_{\underline{t}}^{\underline{t}+T} \Phi'^{\top}\left(X(\tau), \hat{\theta}(\tau)\right)\Phi'\left(X(\tau), \hat{\theta}(\tau)\right)d\tau \leq \varphi_2 I_p$. In this case, it can be shown that $\beta_1 > 0$ (Slotine & Li, 1989, Sec. 4.2).

The following section shows the stability analysis for the developed DNN-based composite adaptive control method over the time-interval $[t_0, \infty) \subseteq \mathbb{R}_{\geq 0}$.

## 3 STABILITY ANALYSIS

DNNs are nonlinear with respect to the weights. Designing adaptive controllers and performing stability analyses for systems that are NIP has historically been a challenging task. A method to address the NIP structure of the uncertainty, especially for DNNs, is to use a first-order Taylor series approximation (Patil et al., 2022a). Let $\tilde{\theta} \triangleq \theta^* - \hat{\theta} \in \mathbb{R}^p$ denote the parameter estimation error. Applying a first-order Taylor series approximation yields

$$\Phi(X, \theta^*) - \Phi\left(X, \hat{\theta}\right) = \Phi'\left(X, \hat{\theta}\right)\tilde{\theta} + \mathcal{O}\left(\left\|\tilde{\theta}\right\|^2\right), \tag{14}$$

where $\mathcal{O}\left(\left\|\tilde{\theta}\right\|^2\right) \in \mathbb{R}^n$ denotes the higher-order terms and $\mathcal{O}(\cdot)$ denotes the asymptotic notation. Substituting (14) into (7) yields the closed-loop error system

$$\dot{r} = \Phi'\left(X, \hat{\theta}\right)\tilde{\theta} + \Delta - e - k_r r, \tag{15}$$

where $\Delta \in \mathbb{R}^n$ is defined as $\Delta \triangleq \mathcal{O}\left(\left\|\tilde{\theta}\right\|^2\right) + \varepsilon(X)$. To facilitate the subsequent analysis, the prediction error $E$ in (10) can be rewritten by adding and subtracting $f$, substituting in (5) and (14), and using the relation $\hat{f} = f - \tilde{f}$, which yields

$$E = \Phi'\left(X, \hat{\theta}\right)\tilde{\theta} - \tilde{f} + \Delta. \tag{16}$$

Taking the time-derivative of $\tilde{\theta}$, substituting in (11), and then applying (16) and the relation $\hat{\theta} = \theta^* - \tilde{\theta}$ yields the parameter estimation error dynamics

$$\begin{aligned} \dot{\tilde{\theta}} = &-\text{proj}\left(\Gamma(t)\left(k_{\hat{\theta}} + \alpha_3 \Phi'^{\top}\left(X, \hat{\theta}\right)\Phi'\left(X, \hat{\theta}\right)\right)\tilde{\theta} + \Gamma(t)\Phi'^{\top}\left(X, \hat{\theta}\right)r \right. \\ &\left. -\alpha_3 \Gamma(t)\Phi'^{\top}\left(X, \hat{\theta}\right)\tilde{f} + \alpha_3 \Gamma(t)\Phi'^{\top}\left(X, \hat{\theta}\right)\Delta - k_{\hat{\theta}}\Gamma(t)\theta^*\right). \end{aligned} \tag{17}$$

Let $z \triangleq \begin{bmatrix} e^{\top} & r^{\top} & \tilde{r}^{\top} & \tilde{f}^{\top} & \tilde{\theta}^{\top} \end{bmatrix}^{\top} \in \mathbb{R}^{4n+p}$ denote the concatenated state. Since the universal function approximation property of the DNN stated in (5) holds only on the compact domain $\Omega$, the

subsequent stability analysis requires ensuring $X(t) \in \Omega$ for all $t \in [t_0, \infty)$. This is achieved by yielding a stability result which constrains $z$ in a compact domain. Consider the compact domain $\mathcal{D} \triangleq \{\zeta \in \mathbb{R}^{4n+p} : \|\zeta\| \leq \chi\}$ in which $z$ is supposed to lie. To facilitate the stability analysis, let $V : \mathbb{R}^{4n+p} \to \mathbb{R}_{\geq 0}$ be the candidate Lyapunov function defined as

$$V(z) = \frac{1}{2}e^\top e + \frac{1}{2}r^\top r + \frac{1}{2}\tilde{r}^\top \tilde{r} + \frac{1}{2}\tilde{f}^\top \tilde{f} + \frac{1}{2}\tilde{\theta}^\top \Gamma^{-1}(t)\tilde{\theta}, \tag{18}$$

which satisfies the inequality

$$\lambda_1 \|z\|^2 \leq V(z) \leq \lambda_2 \|z\|^2, \tag{19}$$

where $\lambda_1 \triangleq \min\{\frac{1}{2}, \frac{1}{2\varkappa_0}\} \in \mathbb{R}_{>0}$ and $\lambda_2 \triangleq \max\{\frac{1}{2}, \frac{1}{2\varkappa_1}\} \in \mathbb{R}_{>0}$. Taking the time-derivative of $V(z)$, substituting in (3), (9), (12), (15), and (17), applying the property of projection operators $-\tilde{\theta}^\top \Gamma^{-1}(t)\mathrm{proj}(\mu) \leq -\tilde{\theta}^\top \Gamma^{-1}(t)\mu$ (Krstic et al., 1995, Lemma E.1.IV), and canceling coupling terms yields

$$\dot{V} \leq -\alpha_1 \|e\|^2 - k_r \|r\|^2 - \alpha_2 \|\tilde{r}\|^2 - k_f \left\|\tilde{f}\right\|^2 - \left(k_{\hat{\theta}} + \frac{\beta(t)}{2\varkappa_0}\right)\left\|\tilde{\theta}\right\|^2 + r^\top \Delta + \tilde{f}^\top \dot{f}$$

$$- \left(\alpha_3 - \frac{1}{2}\right)\tilde{\theta}^\top \Phi'^\top \left(X, \hat{\theta}\right) \Phi'\left(X, \hat{\theta}\right)\tilde{\theta} + \alpha_3 \tilde{\theta}^\top \Phi'^\top \left(X, \hat{\theta}\right)\left(\tilde{f} - \Delta\right) + k_{\hat{\theta}}\tilde{\theta}^\top \theta^*. \tag{20}$$

Due to Assumption 2 and the use of the projection operator, $\|\theta^*\|, \left\|\hat{\theta}\right\| \leq \bar{\theta}$; hence, $\left\|\tilde{\theta}\right\| \leq 2\bar{\theta}$. Additionally, since $f$ and $\Phi$ are continuously differentiable, the bounds $\|\Delta\| \leq \gamma_1$, $\left\|\dot{f}\right\| \leq \gamma_2$, and $\left\|\Phi'\left(X, \hat{\theta}\right)\right\|_F \leq \gamma_3$ hold for all $z \in \mathcal{D}$, where $\gamma_1, \gamma_2, \gamma_3 \in \mathbb{R}_{>0}$ denote bounding constants. Therefore, using Young's inequality yields the bounds $r^\top \Delta \leq \frac{\gamma_1}{2}\|r\|^2 + \frac{\gamma_1}{2}$, $\tilde{f}^\top \dot{f} \leq \frac{\gamma_2}{2}\left\|\tilde{f}\right\|^2 + \frac{\gamma_2}{2}$, $k_{\hat{\theta}}\tilde{\theta}^\top \theta^* \leq \frac{k_{\hat{\theta}}}{2}\left\|\tilde{\theta}\right\|^2 + \frac{k_{\hat{\theta}}}{2}\bar{\theta}^2$, and $\alpha_3 \tilde{\theta}^\top \Phi'^\top \left(X, \hat{\theta}\right)\left(\tilde{f} - \Delta\right) \leq \alpha_3\gamma_3 \left\|\tilde{\theta}\right\|^2 + \frac{\alpha_3\gamma_3}{2}\left\|\tilde{f}\right\|^2 + \frac{\alpha_3\gamma_3\gamma_1^2}{2}$. As a result, $\dot{V}$ can further be upper-bounded as

$$\dot{V} \leq -\lambda_3 \|z\|^2 + c - \left(\alpha_3 - \frac{1}{2}\right)\tilde{\theta}^\top \Phi'^\top \left(X, \hat{\theta}\right) \Phi'\left(X, \hat{\theta}\right)\tilde{\theta}, \tag{21}$$

for all $z \in \mathcal{D}$, where $\lambda_3 \triangleq \min\{\alpha_1, k_r - \frac{\gamma_1}{2}, \alpha_2, k_f - \frac{\gamma_2 + \alpha_3\gamma_3}{2}, \frac{k_{\hat{\theta}}}{2} + \frac{\beta_1}{2\varkappa_0} - \alpha_3\gamma_3\} \in \mathbb{R}$ and $c \triangleq \frac{\gamma_1 + \gamma_2 + k_{\hat{\theta}}\bar{\theta}^2 + \alpha_3\gamma_3\gamma_1^2}{2} \in \mathbb{R}_{>0}$. To facilitate the subsequent analysis, the following gain condition is introduced

$$\min\left\{\lambda_3, \alpha_3 - \frac{1}{2}\right\} > 0. \tag{22}$$

Additionally, the set $\mathcal{S} \triangleq \{\zeta \in \mathbb{R}^{4n+p} : \|\zeta\| \leq \sqrt{\frac{\lambda_1}{\lambda_2}\chi^2 - \frac{c}{\lambda_3}}\}$ is defined to initialize $z$ in the subsequent analysis, where it is shown that if $z(t_0) \in \mathcal{S} \subset \mathcal{D}$ then $z(t)$ is UUB and does not escape $\mathcal{D}$. The following theorem states the main result of this paper.

**Theorem 2.** *Let Assumptions 1 and 2 and the gain condition in (22) hold, and let $\chi > \sqrt{\frac{\lambda_2 c}{\lambda_1 \lambda_3}}$. Then, for the system in (1), the DNN-based controller in (6) and the composite adaptation law in (11) ensure $z$ is UUB in the sense that $\|z(t)\| \leq \sqrt{\frac{\lambda_2}{\lambda_1}\|z(t_0)\|^2 e^{-\frac{\lambda_3}{\lambda_2}(t-t_0)} + \frac{\lambda_2 c}{\lambda_1 \lambda_3}\left(1 - e^{-\frac{\lambda_3}{\lambda_2}(t-t_0)}\right)}$ for all $t \in [t_0, \infty)$, provided that $\|z(t_0)\| \in \mathcal{S}$.*

*Proof.* See Appendix A. □

*Remark* 3. Since $\lambda_3 = \min\{\alpha_1, k_r - \frac{\gamma_1}{2}, \alpha_2, k_f - \frac{\gamma_2 + \alpha_3\gamma_3}{2}, \frac{k_{\hat{\theta}}}{2} + \frac{\beta_1}{2\varkappa_0} - \alpha_3\gamma_3\}$, the gains $\alpha_1, \alpha_2, \alpha_3, k_r$, and $k_f$ can be selected to be sufficiently high such that $\lambda_3 = \frac{k_{\hat{\theta}}}{2} + \frac{\beta_1}{2\varkappa_0} - \alpha_3\gamma_3$. Since $\beta_1$ is positive under the PE condition as mentioned in Remark 1, a larger value for $\lambda_3$ is obtained, which implies faster exponential convergence to a smaller neighborhood of the origin. When the PE condition does not hold, the gain $k_{\hat{\theta}}$, which is based on the sigma modification technique in (Ioannou and Sun, 1996, Sec. 8.4.1), helps achieve the UUB stability result. However, selecting a high value for $k_{\hat{\theta}}$ can deteriorate tracking and parameter estimation performance since it yields a higher value for $c$.

Table 1: Robot Manipulator Performance Comparison

| Adaptation Law | $\|e\|_{\text{RMS}}$ (deg) | $\|u\|_{\text{RMS}}$ (Nm) | Function error on-trajectory $(\text{rad}/\text{s}^2)$ | Mean function error on test data $(\text{rad}/\text{s}^2)$ |
|---|---|---|---|---|
| Tracking Error-Based | 0.629 | 10.100 | 0.430 | 1.215 |
| Composite | 0.308 | 7.962 | 0.131 | 0.260 |
| Observer-Based | 0.310 | 10.612 | 0.204 | N/A |
| Nonlinear PD | 3.142 | 6.642 | N/A | N/A |
| Nonlinear MPC | 1.101 | 8.275 | N/A | N/A |

## 4 SIMULATIONS

To demonstrate the performance of the developed method, comparative simulations are performed on two different systems, i.e. a two-link manipulator and a UUV.

### 4.1 TWO LINK MANIPULATOR

To demonstrate the performance of the developed composite adaptive Lb-DNN, comparative simulations are performed on the two-link robot manipulator model (see Appendix D.1.1 for the dynamics) in (de Queiroz, Hu, Dawson, Burg, and Donepudi, 1997) for 100 seconds. Baseline methods used for comparison include DNN-based adaptive controller with tracking error-based adaptation law developed in (Patil et al., 2022a), an observer-based disturbance rejection controller (Han, 2009) (i.e., $u = g^+(x, \dot{x}) \left( \ddot{x}_d - (\alpha_1 + k_r) r + (\alpha_1^2 - 1) e - \hat{f} \right)$), a nonlinear proportional-derivative (PD) controller $u = g^+(x, \dot{x}) \left( \ddot{x}_d - (\alpha_1 + k_r) r + (\alpha_1^2 - 1) e \right)$, and nonlinear model predictive control (MPC) (see Appendix D.1.1 for more details on baseline control methods). The comparative simulation is performed using a fully-connected DNN with 5 hidden layers and 5 neurons in each layer with hyperbolic tangent activation functions (see Appendix D.1.2 for ablation study). The DNN weights are initialized randomly from the distribution $U(-0.5, 0.5)$. For a realistic simulation, an additive white Gaussian (AWG) measurement noise with a signal-to-noise ratio of 50 dB is considered in all state measurements.

To evaluate the tracking and drift compensation performance of the developed and baseline methods, the root mean square (RMS) values of the tracking error norm, denoted by $\|e\|_{\text{RMS}}$, and function approximation error along the trajectory are calculated in the steady state (i.e., in the interval [50,100] seconds). The corresponding values are provided in Table 1. Since the trajectory explored by the system essentially acts as a training dataset for the DNNs, the RMS function approximation error does not indicate whether the DNN model is overfit and how well the model generalizes over unexplored data. Thus, to evaluate the performance of the DNN beyond the trajectory, a test dataset involving 100 random datapoints with values selected from the distribution $U(-0.25, 0.25)$ is constructed, and the mean $\left\| f(x, \dot{x}) - \Phi\left(X, \hat{\theta}\right) \right\|$ across all points in the dataset is evaluated. The value of the mean $\left\| f(x, \dot{x}) - \Phi\left(X, \hat{\theta}\right) \right\|$ on the test dataset at the end of each simulation (i.e., at $t = 100$ seconds) is then used as a metric in Table 1 for comparing the generalization performance of each method. To evaluate the control effort required by each controller throughout the transient and steady states, the RMS values of the control input norm, $\|u\|$, are calculated in the time-interval [0,100] seconds and provided in Table 1 for each method. As evident from Table 1 and Figure 1, the developed composite adaptive Lb-DNN significantly improved the tracking performance compared to tracking error-based adaptive Lb-DNN, nonlinear PD, and nonlinear MPC with comparable control effort and approximately 50%, 90%, and 70% improvements in $\|e\|_{\text{RMS}}$, respectively. The tracking error-based adaptive Lb-DNN exhibited more chattering in the control input due to measurement noise, which might be because the update law involved a constant high adaptation gain. In contrast, the composite update law has a decreasing gain due to the least squares approach which mitigates noise amplification resulting from the adaptation gain. Additionally, the tracking performance with the observer-based disturbance rejection method is comparable to the developed method, which is expected because the developed method also used the observer-based estimate $\hat{f}$ to formulate the prediction error. However, notice the increased control effort due to

Table 2: UUV Performance Comparison

|  | $e$-based | Composite | Observer-based | NMPC | NPD |
|---|---|---|---|---|---|
| RMS position tracking error norm (m) | 0.201 | 0.152 | 0.158 | 0.254 | 0.186 |
| RMS angular tracking error norm (rad) | 0.037 | 0.012 | 0.024 | 0.054 | 0.028 |
| RMS linear control input norm (N) | 0.069 | 0.065 | 0.126 | 0.128 | 0.067 |
| RMS angular control input norm (Nm) | 0.041 | 0.035 | 0.072 | 0.036 | 0.032 |
| RMS linear dynamics estimation error norm ($\mathrm{m/s^2}$) | 4.370 | 2.585 | 19.423 | N/A | N/A |
| RMS angular dynamics estimation error norm ($\mathrm{rad/s^2}$) | 2.058 | 1.408 | 3.021 | N/A | N/A |

large overshoots in the controller resulting from high gains in the state-derivative observer in (8). Although the developed composite adaptation law also used the state-derivative estimates generated by the high-gain observer, the state-derivative estimates are not directly used in the control input. Using the state-derivative estimates in the adaptation law did not cause as large overshoots in the control input because the adaptation law involves an integrator that effectively acts as a low pass filter on any overshoots in the state-derivative estimates. Furthermore, note that the observer-based controller only provides instantaneous estimates of $f$, due to the lack of a model. Hence, it cannot be generalized for off-trajectory points, thus not achieving the system identification objective. Additionally, despite the fact that nonlinear MPC uses model knowledge, the developed method achieved improved tracking with reduced control effort compared to nonlinear MPC.

The evolution of the mean function approximation error on the aforementioned test dataset is shown on the right in Figure 1. The mean function approximation error with the composite method on the test dataset initially overshot followed by oscillatory behavior during the initial 10 seconds. Such a behavior is expected since the combined system goes through the initial transients, and the online data in the first few seconds based on which the DNN has learned is limited. However, after 10 s, the composite adaptation law exhibited a consistent decrease in the mean function approximation error, unlike the tracking error-based adaptation law. As a result, the composite adaptation law achieved 72.04% improvement in the final value of mean function approximation error.

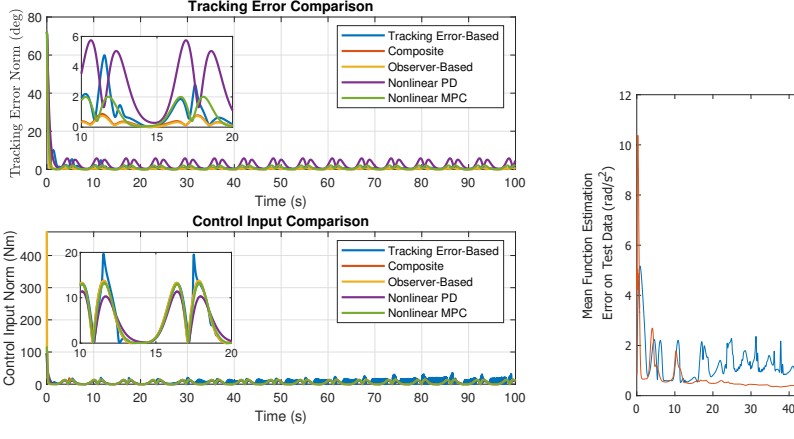

Figure 1: Left: Comparative plots of the tracking error norm and control input norm along the trajectory with the developed and baseline controllers. A zoomed view during the time interval $[10, 20]$ is added in each subplot for visual clarity. Right: Comparative plots of the mean of function estimation error norm $\left\| f(x, \dot{x}) - \Phi\left(X, \hat{\theta}\right) \right\|$ using tracking error-based adaptation and composite adaptation on the test dataset. (See Fig. 4-5 in Appendix D for larger plots)

## 4.2 UNMANNED UNDERWATER VEHICLE

Comparative simulation results are also provided for the UUV system (see Appendix D.2 for dynamics) from (Fischer, Hughes, Walters, Schwartz, and Dixon, 2014) using the composite adaptive

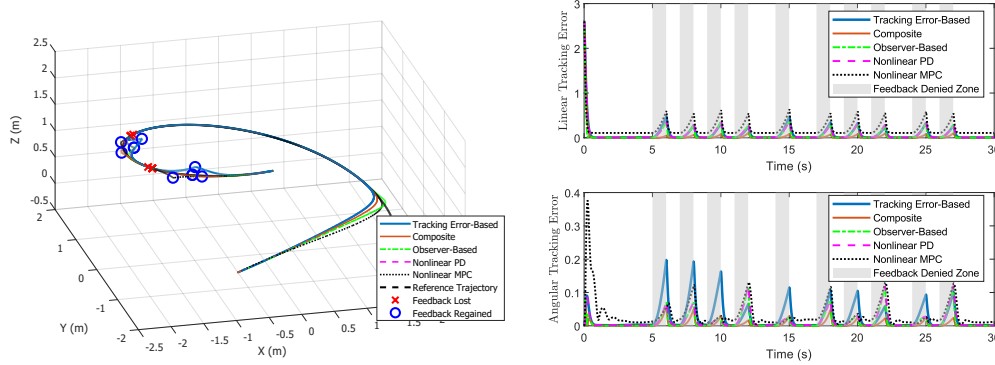

Figure 2: Left: Plot of the position trajectories taken by the UUV using the developed and baseline methods. The plot is restricted to the trajectories from the time-interval $[0, 9]$ seconds for visual clarity. The points where each controller loses feedback are marked by a red $\times$ and points where each controller regains feedback are marked by a blue $\bigcirc$. Right: Comparative plots of the linear tracking error norm (m) and angular tracking error norm (rad) for the UUV. The time intervals corresponding to the feedback denied zones are marked in grey patches. (See Fig. 6-7 in Appendix D for larger plots)

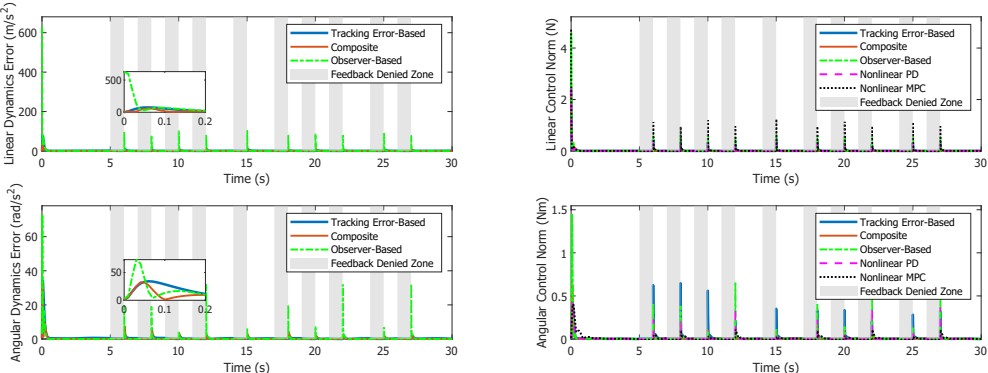

Figure 3: Left: Comparative plots of the estimation error norm (i.e., $\left\| f(x, \dot{x}) - \Phi\left(X, \hat{\theta}\right) \right\|$ during feedback availability and $\left\| f(x, \dot{x}) - \Phi\left(X_d(t), \hat{\theta}\right) \right\|$ during loss of feedback) for the UUV with the developed and baseline methods. Right: Comparative plots of the linear and angular control input norms for the UUV. Zoomed views during the time interval $[0, 0.2]$ seconds are added for visual clarity. (See 8-9 in Appendix D for larger plots)

Lb-DNN under intermittent loss of feedback, with the same baselines as in Subsection 4.1. Since the DNN identifies the system dynamics, the identified DNN could be used to predict the uncertainty when the state feedback is intermittently lost. Let $i \in \mathbb{Z}_{\geq 0}$ denote the time index such that the state feedback is available in the time interval $[t_{2i}, t_{2i+1})$ and unavailable in the time interval $[t_{2i+1}, t_{2i+2})$ for all $i \in \mathbb{Z}_{\geq 0}$. During the time interval $[t_{2i}, t_{2i+1})$, when the feedback is available, the control and adaptation laws in (6) and (11) are used for all $i \in \mathbb{Z}_{\geq 0}$. However, during the time interval $[t_{2i+1}, t_{2i+2})$ when the state feedback is unavailable, an open-loop controller is developed as $u = g^+\left(x_d(t), \dot{x}_d(t)\right)\left(\ddot{x}_d(t) - \Phi\left(X_d(t), \hat{\theta}(t_{2i+1})\right)\right)$. The reader is referred to Appendix E for sufficient dwell-time conditions and stability analysis under which the system can stably operate under intermittent loss of feedback. For both the composite and tracking error-based adaptive Lb-DNN methods, a fully-connected DNN with 5 hidden layers with 5 neurons in each layer with hyperbolic tangent activation function was used. During feedback unavailability, the observer-based disturbance rejection and nonlinear PD controllers were designed to be $u = g^+\left(x_d(t), \dot{x}_d(t)\right)\ddot{x}_d(t)$, with $\hat{\dot{x}}, \hat{\dot{f}} = 0$ for the observer-based method, and the nonlinear MPC was designed using model

predictions propagated forward over the horizon treating $x_d$ as the current state. To simulate the performance of the system under intermittent loss of feedback, each simulation was performed for 30 seconds where the feedback was made unavailable for the time intervals $[5, 6)$, $[7, 8)$, $[9, 10)$, $[11, 12)$, $[14, 15)$, $[17, 18)$, $[19, 20)$, $[21, 22)$, $[24, 25)$, and $[26, 27)$ seconds, respectively. For a realistic simulation, an AWG measurement noise with a signal-to-noise ratio of 50 dB is considered in all state measurements.

Figure 2, on the left, shows the reference trajectory and the actual trajectories taken by the UUV using each controller. On the right, Figure 2 shows comparative plots of linear tracking error and angular tracking error norms. The developed method outperforms the baseline methods in tracking the reference trajectory, especially during the loss of feedback as also evident from Table 2. Additionally, Figure 3 shows the comparative plot of the linear and angular dynamics (function) estimation error norms on the left and control input norms on the right. Since tracking error-based adaptation does not involve guarantees on parameter estimation, the resulting predictions quickly diverge during absence of feedback due to model identification errors. However, the composite adaptive Lb-DNN controller can better predict and compensate for the drift dynamics under the absence of state feedback. Additionally, when feedback is available, the state-derivative observer-based approach can yield a tracking performance comparable to the DNN-based controllers since it essentially involves a robust high-gain approach. However, the tracking performance degrades significantly in the absence of feedback using the observer-based approach when compared to the composite adaptive Lb-DNN controller. Furthermore, the observer-based approach requires significantly higher control effort as compared to both of the DNN-based adaptive controllers due to reasons discussed in Subsection 4.1. Additionally, despite the fact that nonlinear MPC uses model knowledge, the developed method achieved improved tracking with reduced control effort compared to nonlinear MPC.

## 5 CONCLUSION

A composite adaptive Lb-DNN is developed for simultaneous online system identification and control, using the Jacobian of the DNN, the tracking error, and a prediction error based on a novel formulation using a dynamic state-derivative observer. A Lyapunov-based stability analysis guarantees the tracking, observer, and parameter estimation errors are UUB, with tighter bounds on these errors when the DNN's Jacobian satisfies the PE condition. Comparative simulation results demonstrate a significant improvement in tracking, function estimation and generalization capabilities with the developed method in comparison to the tracking error-based Lb-DNN in (Patil et al., 2022a) and observer-based disturbance rejection controller as baseline methods.

## 6 LIMITATIONS AND SCOPE FOR FUTURE WORK

The persistence of excitation condition for identifying the parameters is restrictive as there are challenges in verifying it online. For linear regression, recent developments in the adaptive control literature such as (Chowdhary, Yucelen, Mühlegg, and Johnson, 2013b; Ortega, Aranovskiy, Pyrkin, Astolfi, and Bobtsov, 2021; Pan and Yu, 2018; Parikh, Kamalapurkar, and Dixon, 2019; Roy, Bhasin, and Kar, 2018) provide parameter estimation methods that guarantee parameter convergence under excitation conditions weaker than PE. All of these methods involve some form of regression extension by storing the history of the regression in memory over an interval of time. However, these methods are restricted to linear regression and have not been explored for NIP models such as DNNs yet. Thus, insights from this paper may be used in future work to develop adaptation laws for DNNs that yield parameter estimation guarantees under excitation conditions weaker than PE. Furthermore, extensions of the developed online system identification approach in optimization-based control paradigms such as MPC and reinforcement learning can be explored. Moreover, future research efforts can also investigate how to combine the developed method with control barrier functions to satisfy state and input constraints.

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

APPENDICES

# A PROOFS

## A.1 THEOREM PROOF

*Proof of Theorem 1.* Consider the candidate Lyapunov function in (18). Then, using (19) and (21), when the gain condition in (22) is satisfied, $\dot{V}$ can be upper-bounded as

$$\dot{V} \leq -\frac{\lambda_3}{\lambda_2} V + c, \tag{23}$$

for all $z \in \mathcal{D}$. Solving the differential inequality in (23) over the time-interval $[t_0, \infty)$ yields

$$V(z(t)) \leq V(z(t_0)) e^{-\frac{\lambda_3}{\lambda_2}(t-t_0)} + \frac{\lambda_2 c}{\lambda_3}\left(1 - e^{-\frac{\lambda_3}{\lambda_2}(t-t_0)}\right), \tag{24}$$

for all $z \in \mathcal{D}$. Then applying (19) to (24) yields

$$\|z(t)\| \leq \sqrt{\frac{\lambda_2}{\lambda_1}\|z(t_0)\|^2 e^{-\frac{\lambda_3}{\lambda_2}(t-t_0)} + \frac{\lambda_2 c}{\lambda_1 \lambda_3}\left(1 - e^{-\frac{\lambda_3}{\lambda_2}(t-t_0)}\right)}, \tag{25}$$

for all $z \in \mathcal{D}$. To ensure $z(t) \in \mathcal{D}$ for all $t \in \mathbb{R}_{\geq 0}$, further upper-bounding the right hand side of (25) yields $\|z(t)\| \leq \sqrt{\frac{\lambda_2}{\lambda_1}\|z(t_0)\|^2 + \frac{\lambda_2 c}{\lambda_1 \lambda_3}}$ for all $t \in \mathbb{R}_{\geq 0}$. Since $\mathcal{D} = \left\{\zeta \in \mathbb{R}^{4n+p} : \|\zeta\| \leq \chi\right\}$, $z(t) \in \mathcal{D}$ always holds if $\sqrt{\frac{\lambda_2}{\lambda_1}\|z(t_0)\|^2 + \frac{\lambda_2 c}{\lambda_1 \lambda_3}} \leq \chi$, which is guaranteed if $\|z(t_0)\| \leq \sqrt{\frac{\lambda_1}{\lambda_2}\chi^2 - \frac{c}{\lambda_3}}$, i.e., $z(t_0) \in \mathcal{S}$. Thus, the trajectories of $z$ do not escape $\mathcal{D}$ if $z$ is initialized in $\mathcal{S}$. Since $\|z\| \leq \chi$ implies $\|e\|, \|r\| \leq \chi$, the following relation holds: $\|X\| \leq \|x\| + \|\dot{x}\| \leq \|e + x_d\| + \|r - \alpha_1 e + \dot{x}_d\| \leq (\alpha_1 + 2)\chi + \overline{x_d} + \overline{\dot{x}_d}$. Thus, based on the definition of the set $\Omega$, $X(t) \in \Omega$ for all $t \in [t_0, \infty)$. Furthermore, for the feasibility of initial conditions, $\mathcal{S}$ is required to be non-empty, which is ensured by selecting $\chi > \sqrt{\frac{\lambda_2 c}{\lambda_1 \lambda_3}}$. $\qquad\square$

## A.2 ROBUSTNESS OF DYNAMIC STATE-DERIVATIVE ESTIMATOR TO SENSOR NOISE

Consider noisy state measurement for $r$ given by $r_n = r + \delta(t)$, where $\delta(t) \in \mathbb{R}^n$ is a bounded noise assumed to satisfy $\|\delta(t)\| \leq \bar{\delta}$ with the bounding constant $\bar{\delta} \in \mathbb{R}_{>0}$. Replacing the $r$ term in (9) with the noisy measurement $r_n$ yields

$$\dot{\tilde{r}} = \tilde{f} - \alpha_2 \tilde{r} - \alpha_2 \delta(t) \tag{26}$$

$$\dot{\tilde{f}} = \dot{f} - k_f \tilde{f} - \tilde{r} - \delta(t). \tag{27}$$

To analyse the noise sensitivity, consider the candidate Lyapunov function $V_n = \frac{1}{2}\tilde{r}^\top \tilde{r} + \frac{1}{2}\tilde{f}^\top \tilde{f}$. Taking its time-derivative and substituting (26) and (27) and canceling the cross terms yields

$$\dot{V}_n = -\alpha_2 \|\tilde{r}\|^2 - k_f \left\|\tilde{f}\right\|^2 - \alpha_2 \tilde{r}^\top \delta(t) + \tilde{f}^\top \left(\dot{f} - \delta(t)\right). \tag{28}$$

Using Young's inequality, $-\alpha_2 \tilde{r}^\top \delta(t) \leq \frac{\alpha_2}{2}\|\tilde{r}\|^2 + \frac{\alpha_2}{2}\|\delta(t)\|^2 \leq \frac{\alpha_2}{2}\|\tilde{r}\|^2 + \frac{\alpha_2}{2}\bar{\delta}^2$ and $\tilde{f}^\top\left(\dot{f} - \delta(t)\right) \leq \frac{k_f}{2}\left\|\tilde{f}\right\|^2 + \frac{1}{2k_f}\left\|\dot{f} - \delta(t)\right\|^2 \leq \frac{k_f}{2}\left\|\tilde{f}\right\|^2 + \frac{1}{k_f}\left(\left\|\dot{f}\right\|^2 + \|\delta(t)\|^2\right) \leq \frac{k_f}{2}\left\|\tilde{f}\right\|^2 + \frac{\gamma_2^2 + \bar{\delta}^2}{k_f}$. Substituting these inequalities into (28) yields

$$\dot{V}_n \leq -\frac{\alpha_2}{2}\|\tilde{r}\|^2 - \frac{k_f}{2}\left\|\tilde{f}\right\|^2 + \frac{\gamma_2^2 + \bar{\delta}^2}{k_f} + \frac{\alpha_2}{2}\bar{\delta}^2$$

$$\leq -\lambda_n V_n + c_n, \tag{29}$$

where $\lambda_n \triangleq \frac{1}{2}\min(\alpha_2, k_f)$ and $c_n \triangleq \frac{\gamma_2^2 + \bar{\delta}^2}{k_f} + \frac{\alpha_2}{2}\bar{\delta}^2$. Solving the differential inequality in (29) yields $V_n(t) = V_n(t_0) e^{-\lambda_n(t-t_0)} + \frac{c_n}{\lambda_n}\left(1 - e^{-\lambda_n(t-t_0)}\right)$. Let $z_n \triangleq \begin{bmatrix} \tilde{r}^\top & \tilde{f}^\top \end{bmatrix}^\top$. Using $V_n = \frac{1}{2}\|z_n\|^2$ yields $\|z_n(t)\| \leq \sqrt{\|z_n(t_0)\| e^{-\lambda_n(t-t_0)} + \frac{2c_n}{\lambda_n}\left(1 - e^{-\lambda_n(t-t_0)}\right)}$, implying the error $\tilde{f}$ is UUB with the ultimate bound $\sqrt{\frac{2c_n}{\lambda_n}} = \sqrt{\frac{2}{\lambda_n}\left(\frac{\gamma_2^2 + \bar{\delta}^2}{k_f} + \frac{\alpha_2}{2}\bar{\delta}^2\right)}$. Thus, the state-derivative estimator is robust to noise in the sense the ultimate bound on $\tilde{f}$ grows linearly with $\bar{\delta}$.

## B    RELATED WORK

### B.1    ON NEURAL NETWORK-BASED ADAPTIVE CONTROL

Classical results (Ge, Hang, Lee, and Zhang, 2002; Lewis, Yesildirek, and Liu, 1996b; Lewis, 1996) develop adaptive controllers for neural networks with a single hidden layer, where online updates are performed for the input and output layer weights. In recent results (Joshi et al., 2020; Le et al., 2022a; Sun et al., 2022), adaptive controllers were developed for DNNs. In these results, the outer-layer weights of the DNN are updated in real-time using Lyapunov-based adaptation laws, whereas the inner-layer weights are updated either using iterative batch updates on discrete intervals of time (Joshi et al., 2020; Le et al., 2022a; Sun et al., 2022), or using a modular design (Le et al., 2022a). Since the inner-layer weight updates in (Joshi et al., 2020; Le et al., 2022a; Sun et al., 2022) happen using batch updates, the updates are essentially performed offline. In (Le et al., 2022a), the weights are updated online but the update laws are selected arbitrarily and not by a stability-driven approach. In (Patil et al., 2022a), Lyapunov-based adaptation laws are developed for all layers of a fully-connected DNN (i.e., so-called Lb-DNN methods). Since the control and adaptation laws are derived from a Lyapunov-based stability analysis, the development is guaranteed to ensure stability of the closed-loop system. More recent Lb-DNN results develop Lyapunov-based adaptation laws for more complex architectures, specifically, deep residual networks (ResNets) (Patil, Le, Griffis, and Dixon, 2022b) and long short-term memory (LSTM) networks (Griffis et al., 2023). However, as stated in the manuscript, the updates are based solely on tracking error feedback and are primarily meant to achieve tracking error convergence. These results do not achieve guarantees on parameter estimation and system identification.

### B.2    ON COMPOSITE ADAPTIVE CONTROL

The classical result in (Slotine & Li, 1989) develops adaptive controllers with a composite adaptation law that includes both tracking and prediction errors for nonlinear systems with linear-in-parameters (LIP) uncertainties. The result in (Slotine & Li, 1989) constructs a form of the prediction error using the swapping technique (also known as input or torque filtering), where a low-pass filter is applied on both sides of the dynamics to eliminate the unknown state-derivative term. For a brief illustration of the swapping technique in (Slotine & Li, 1989), consider the system

$$\dot{x} = Y(x)\theta + u,$$

where $Y(x)$ is the regressor, $\theta$ is the vector of unknown parameters, and $u$ is the control input. If the state-derivative $\dot{x}$ could be measured, the system can be expressed in terms of the linear regression equation $\dot{x} - u = Y(x)\theta$. Then the corresponding prediction error with an adaptive estimate $\hat{\theta}$ could be developed as $\epsilon = \dot{x} - u - Y(x)\hat{\theta}$, which can be expressed linearly in terms of the parameter estimation error $\tilde{\theta} = \theta - \hat{\theta}$ as $\epsilon = Y(x)\tilde{\theta}$. However, $\dot{x}$ measurements are typically either unavailable or extremely noisy. To avoid using state-derivative information, (Slotine & Li, 1989) applied a low-pass filter on both sides of the dynamics, which results in the filtered regression

$$e^{-\beta t} * (\dot{x} - u) \quad = \quad \left( e^{-\beta t} * Y(x) \right) \theta,$$

where $*$ denotes the convolutional integral operation (i.e., $a(t) * b(t) = \int_0^t a(t - \tau)b(\tau)d\tau$) and $e^{-\beta t}$ is the impulse response of the low-pass filter with a positive constant decay rate $\beta$. Since $e^{-\beta t} * \dot{x} = x(t) - x(0)e^{-\beta t} + \beta e^{-\beta t} * x$, the filtered regression can be expressed as

$$x(t) - x(0)e^{-\beta t} + \beta e^{-\beta t} * x - e^{-\beta t} * u = \left( e^{-\beta t} * Y(x) \right) \theta,$$

which is implementable without using state-derivative information. The prediction error for the filtered regression can be developed as

$$
\begin{aligned}
\epsilon \quad &= \quad x(t) - x(0)e^{-\beta t} + \beta e^{-\beta t} * x - e^{-\beta t} * u - \left( e^{-\beta t} * Y(x) \right) \hat{\theta} \\
&= \quad \left( e^{-\beta t} * Y(x) \right) \theta - \left( e^{-\beta t} * Y(x) \right) \hat{\theta} \\
&= \quad \left( e^{-\beta t} * Y(x) \right) \tilde{\theta}. \\
&= \quad Y_f \tilde{\theta},
\end{aligned}
$$

where $Y_f = \left(e^{-\beta t} * Y(x)\right)$. Since $\epsilon$ is linear in $\tilde{\theta}$, a composite adaptation law can be developed with a $Y_f^\top \epsilon$ term which would yield negative $\tilde{\theta}$ terms in the corresponding Lyapunov-based stability analysis. However, yielding this form of $\epsilon$ using a filtered regression was possible because the uncertainty $Y(x)\theta$ is linear in terms of $\theta$, which allowed $\theta$ to be separable from $e^{-\beta t} * Y(x)$ in the filtered regression. If $Y(x)\theta$ is replaced by terms nonlinear in $\theta$, such as the DNN-based approximation $\Phi(x, \theta) + \varepsilon(x)$, applying a low pass filter on both sides would yield

$$e^{-\beta t} * (\dot{x} - u) = e^{-\beta t} * (\Phi(x, \theta) + \varepsilon(x)).$$

Notice that $\theta$ is not separable from the convolutional integral in the term $e^{-\beta t} * \Phi(x, \theta)$ since $\Phi$ is nonlinear in $\theta$. As a result, the swapping technique from (Slotine & Li, 1989) does not apply for nonlinear in parameter uncertainties such as DNNs.

Results in (Patre, Mackunis, Johnson, and Dixon, 2010b) introduce a robust integral of the sign of the error (RISE)-based swapping technique to formulate the prediction error and design composite adaptive controllers for LIP uncertainties with additive disturbances. The RISE-based swapping technique is extended in (Patre, Bhasin, Wilcox, and Dixon, 2010a) for NN-based models, but the development is restricted to single-hidden-layer NNs. Extending this for DNNs is mathematically challenging due to their nested NIP structure. Moreover, using RISE-based swapping requires additional RISE-based terms in the control input, which can debilitate the learning performance of the adaptive feedforward term. Notably, the results in (Patre et al., 2010b) and (Patre et al., 2010a) only ensure asymptotic tracking error convergence, and no guarantees are provided on the parameter estimates under the persistence of excitation (PE) condition.

The recent result in (OConnell, Shi, Shi, Azizzadenesheli, Anandkumar, Yue, and Chung, 2022) developed a new learning representation uncertainties involving a composited disturbance given by $f(x, \dot{x}, w) = \phi(x, \dot{x}) a(w)$, where $\phi(\cdot)$ denotes a basis function that is learned using a DNN and $a(w)$ denotes a set of linear parameters accounting for an unknown disturbance time-varying disturbance $w$. Since $f$ is linear in terms of $a$, the composite adaptive approach from (Slotine & Li, 1989) is used to design an adaptation law $\dot{\hat{a}}$ to update the estimates of $a$ given by $\hat{a}$. To obtain a disturbance-invariant representation of $\phi$ using DNNs, a domain adversarially invariant meta-learning (DAIML) algorithm is developed to train the DNN offline. To the best of our knowledge, this is the only existing work using a composite adaptive approach in the context of deep learning-based control. However, since the DNN learning $\phi(x, \dot{x})$ has an NIP structure, the aforementioned challenges apply for constructing a Lyapunov-based online adaptation law.

### B.3 On Alternative Data-Driven Approaches

Besides DNNs, other methods such as Gaussian processes (GPs), Koopman operator, kernel methods have been explored to compensate for system uncertainty (Beckers, Kulic, and Hirche, 2019; Berkenkamp and Schoellig, 2015; Bevanda, Sosnowski, and Hirche, 2021; Chowdhary, Kingravi, How, and Vela, 2013a; Joshi and Chowdhary, 2018; Kingravi, Chowdhary, Vela, and Johnson, 2012; Lederer, Umlauft, and Hirche, 2019; Umlauft and Hirche, 2020). GP models can be updated online and implemented in the control design as a feedforward estimate of nonlinear system uncertainties. In (Beckers et al., 2019; Berkenkamp & Schoellig, 2015; Lederer et al., 2019), safety guarantees are derived for the developed adaptive GP architectures and overall control designs. The use of a probabilistic model provides a measure of confidence in the GP estimates. Similar to Lb-DNN-based control, the use of adaptive GPs is motivated by integrating the data-driven estimation capabilities of modern machine learning models with the stability guarantees, online learning capabilities, and robustness to real-time disturbances of adaptive control techniques.

## C Deep Neural Network Model

For simplicity in the illustration, a fully-connected DNN will be described here. The following control and adaptation law development can be generalized for any network architecture $\Phi$ with a corresponding Jacobian $\Phi'$. The reader is referred to (Patil et al., 2022b) and (Griffis et al., 2023) for extending the subsequent development to ResNets and LSTMs, respectively. Given some matrix $A \triangleq [a_{i,j}] \in \mathbb{R}^{n \times m}$, where $a_{i,j}$ denotes the element in the $i^{th}$ row and $j^{th}$ column of $A$, the vectorization operator is defined as $\text{vec}(A) \triangleq [a_{1,1}, \dots, a_{n,1}, \dots, a_{1,m}, \dots, a_{n,m}]^\top \in \mathbb{R}^{nm}$. Let

$\sigma \in \mathbb{R}^{L_{\text{in}}}$ denote the DNN input with size $L_{\text{in}} \in \mathbb{Z}_{>0}$, and $\theta \in \mathbb{R}^p$ denote the vector of DNN parameters (i.e., weights and bias terms) with size $p \in \mathbb{Z}_{>0}$. Then, a fully-connected feedforward DNN $\Phi(\sigma, \theta)$ with output size $L_{\text{out}} \in \mathbb{Z}_{>0}$ is defined using a recursive relation $\Phi_j \in \mathbb{R}^{L_{j+1}}$ given by

$$\Phi_j \triangleq \begin{cases} V_j^\top \phi_j(\Phi_{j-1}), & j \in \{1, \ldots, k\}, \\ V_j^\top \sigma_a, & j = 0, \end{cases} \tag{30}$$

where $\Phi(\sigma, \theta) = \Phi_k$, and $\sigma_a \triangleq \begin{bmatrix} \sigma^\top & 1 \end{bmatrix}^\top$ denotes the augmented input that accounts for the bias terms, $k \in \mathbb{Z}_{>0}$ denotes the total number of hidden layers, $V_j \in \mathbb{R}^{L_j \times L_{j+1}}$ denotes the matrix of weights and biases, $L_j \in \mathbb{Z}_{>0}$ denotes the number of nodes in the $j^{\text{th}}$ layer for all $j \in \{0, \ldots, k\}$ with $L_0 \triangleq L_{\text{in}} + 1$ and $L_{k+1} = L_{\text{out}}$. The vector of smooth activation functions is denoted by $\phi_j : \mathbb{R}^{L_j} \to \mathbb{R}^{L_j}$ for all $j \in \{1, \ldots, k\}$. If the DNN involves multiple types of activation functions at each layer, then $\phi_j$ may be represented as $\phi_j \triangleq \begin{bmatrix} \varsigma_{j,1} & \cdots & \varsigma_{j,L_j-1} & 1 \end{bmatrix}^\top$, where $\varsigma_{j,i} : \mathbb{R} \to \mathbb{R}$ denotes the activation function at the $i^{\text{th}}$ node of the $j^{\text{th}}$ layer. For the DNN architecture in (30), the vector of DNN weights is $\theta \triangleq \begin{bmatrix} \text{vec}(V_0)^\top & \cdots & \text{vec}(V_k)^\top \end{bmatrix}^\top$ with size $p = \Sigma_{j=0}^k L_j L_{j+1}$. The Jacobian of the activation function vector at the $j^{\text{th}}$ layer is denoted by $\phi_j' : \mathbb{R}^{L_j} \to \mathbb{R}^{L_j \times L_j}$, and $\phi_j'(y) \triangleq \frac{\partial}{\partial z} \phi_j(z)\big|_{z=y}, \forall y \in \mathbb{R}^{L_j}$. Let the Jacobian of the DNN with respect to the weights be denoted by $\Phi'(\sigma, \theta) \triangleq \frac{\partial}{\partial \theta} \Phi(\sigma, \theta)$, which can be represented using $\Phi'(\sigma, \theta) = \begin{bmatrix} \Phi_0', & \Phi_1', & \ldots, & \Phi_k' \end{bmatrix}$, where $\Phi_j' \triangleq \frac{\partial}{\partial \text{vec}(V_j)} \Phi(\sigma, \theta)$ for all $j \in \{0, \ldots, k\}$. Then, using (30) and the property $\frac{\partial}{\partial \text{vec}(B)} \text{vec}(ABC) = C^\top \otimes A$ yields

$$\Phi_0' = \left( \prod_{l=1}^{\curvearrowleft k} V_l^\top \phi_l'(\Phi_{l-1}) \right) (I_{L_1} \otimes \sigma_a^\top), \tag{31}$$

and

$$\Phi_j' = \left( \prod_{l=j+1}^{\curvearrowleft k} V_l^\top \phi_l'(\Phi_{l-1}) \right) \left( I_{L_{j+1}} \otimes \phi_j^\top(\Phi_{j-1}) \right), \tag{32}$$

for all $j \in \{1, \ldots, k\}$. In (31) and (32), the notation $\overset{\curvearrowleft}{\prod}$ denotes the right-to-left matrix product operation, i.e., $\prod_{p=1}^{\curvearrowleft m} A_p = A_m \ldots A_2 A_1$ and $\prod_{p=a}^{\curvearrowleft m} A_p = I$ if $a > m$, and $\otimes$ denotes the Kronecker product.

## D  MORE SIMULATION RESULT DETAILS

All simulations were performed in MATLAB on a desktop with 64 GB RAM and 13th Gen Intel Core i9-13900 @2.00 GHz processor.

### D.1  TWO LINK MANIPULATOR

#### D.1.1  DYNAMIC MODEL

The two-link robot manipulator was modeled by the uncertain Euler-Lagrange dynamics

$$M(x)\ddot{x} + C(x, \dot{x})\dot{x} + F\dot{x} = u, \tag{33}$$

where $x \triangleq [x_1, x_2]^\top \in \mathbb{R}^2$, $\dot{x} \in \mathbb{R}^2$, and $\ddot{x} \in \mathbb{R}^2$ denote the vector of angular position, velocity, and acceleration of joints, respectively, $M(x) \in \mathbb{R}^{2 \times 2}$ represents the inertia matrix, $C(x, \dot{x}) \in \mathbb{R}^{2 \times 2}$ represents the centripetal-Coriolis matrix, $F \in \mathbb{R}^{2 \times 2}$ represents friction effects, and $u \in \mathbb{R}^2$ denotes the torque inputs. In (33), the dynamics were modeled as (de Queiroz et al., 1997)

$$M(x) = \begin{bmatrix} p_1 + 2p_3 c_2, & p_2 + p_3 c_2 \\ p_2 + p_3 c_2, & p_2 \end{bmatrix}, \tag{34}$$

$$C\left(x, \dot{x}\right) = \left[ \begin{array}{cc} -p_3 s_2 \dot{x}_2, & -p_3 s_2 \left(\dot{x}_1 + \dot{x}_2\right) \\ p_3 s_2 \dot{x}_1, & 0 \end{array} \right], \tag{35}$$

$$F = \left[ \begin{array}{cc} f_1, & 0 \\ 0, & f_2 \end{array} \right], \tag{36}$$

where the short-hand notations $c_2$ and $s_2$ are defined as $c_2 \triangleq \cos\left(x_2\right)$ and $s_2 \triangleq \sin\left(x_2\right)$, respectively. The nominal parameters of the two-link robot model in (34)–(36) were $p_1 = 3.473\,\mathrm{kg \cdot m^2}$, $p_2 = 0.196\,\mathrm{kg \cdot m^2}$, $p_3 = 0.242\,\mathrm{kg \cdot m^2}$, $f_1 = 5.3\,\mathrm{Nm \cdot sec}$, and $f_2 = 1.1\,\mathrm{Nm \cdot sec}$. The two-link manipulator dynamics can be expressed using Eq. (1) from the manuscript with $f(x, \dot{x}) = -M^{-1}(x)\left(C\left(x, \dot{x}\right)\dot{x} + F\dot{x}\right)$ and $g(x, \dot{x}) = M^{-1}(x)$. The gains are selected as $\alpha_1 = 5$, $\alpha_2 = 10$, $\alpha_3 = 20$, $\Gamma(0) = I$, $k_r = 5$, $k_f = 10$, $k_{\hat{\theta}} = 0.0001$, $\beta_0 = 10$, and $\varkappa_0 = 2$. The states are initialized as $x(0) = [1, -1]^\top$ rad and $\dot{x}(0) = [0, 0]^\top$ rad/s, the initial parameter estimate $\hat{\theta}(0)$ is selected from the uniform distribution $U(-0.5, 0.5)$, and the desired trajectory is $x_d = 0.25 \exp(-\sin(t))[\sin(t), \cos(t)]^\top$ rad. The weights are randomly initialized from the distribution $U(-0.5, 0.5)$. Baseline methods used for comparison include DNN-based adaptive controller with tracking error-based adaptation law developed in (Patil et al., 2022a), an observer-based disturbance rejection controller (Han, 2009) (i.e., $u = g^+(x, \dot{x})\left(\ddot{x}_d - (\alpha_1 + k_r)\,r + \left(\alpha_1^2 - 1\right)e - \hat{f}\right)$), a nonlinear proportional-derivative (PD) controller $u = g^+(x, \dot{x})\left(\ddot{x}_d - (\alpha_1 + k_r)\,r + \left(\alpha_1^2 - 1\right)e\right)$, and nonlinear model predictive control (MPC) The baseline DNN-based adaptive controller uses the tracking error-based adaptation law given by (Patil et al., 2022a)

$$\dot{\hat{\theta}} = \mathrm{proj}\left(-k_{\hat{\theta}}\Gamma(t_0)\hat{\theta} + \Gamma(t_0)\Phi'^\top\left(X, \hat{\theta}\right)r\right)$$

with a constant $\Gamma$ (unlike the developed method which uses a time-varying $\Gamma$), where it was selected as $\Gamma = I$. For a fair comparison, the set of gains common to the developed and baseline methods were selected to be exactly the same. The nonlinear MPC was designed to minimize the cost

$$\begin{aligned} J(e(t_k), r(t_k), u(t_k)) &= \sum_{i=1}^{N} \left(e(t_{k+i})^\top Q_e e(t_{k+i})\right. \\ &\quad \left. + r(t_{k+i})^\top Q_r r(t_{k+i}) + u(t_{k+i})^\top R u(t_{k+i})\right) \end{aligned}$$

subjected to the model dynamics discretized using Euler's method with a step size of 0.01 seconds. The controller was implemented using MATLAB's fmincon optimizer with $Q_e = I$, $Q_r = I$, $R = 0.0001I$ and a prediction horizon of $N = 5$ steps and bounded control input search space with upper and lower bounds of 50 and -50, respectively, for every control input.

An ablation study is performed to demonstrate the performance of the developed method for various DNN architectures mentioned in Table 3. The same set of gains are used and the weights are randomly initialized from the distribution $U(-0.5, 0.5)$. As evident from the percentage decrease in Table 3, the developed composite adaptation law significantly improves the tracking, drift compensation, and generalization performance of the DNN across all DNN architecture with a comparable control effort. Notably, although all DNN architectures yielded acceptable performance (i.e., with $\|e\|_{\mathrm{RMS}}$ less than 0.5 deg) with the developed composite adaptive Lb-DNN controller, no conclusive trend was obtained to comment on the selection of appropriate size for the DNN for this application. Importantly, using DNN of a greater size did not affect the control effort.

## D.2 UUV System

The simulations were performed on an unmanned underwater vehicle (UUV) system that can be modeled as (Fischer et al., 2014)

$$\ddot{x} = -\overline{M}^{-1}(x)\left(\overline{C}\left(x, \dot{x}, \nu\right)\dot{x} + \overline{D}\left(x, \nu\right)\dot{x} + \overline{G}\left(x\right)\right) + \overline{M}^{-1}(x)\tau_n, \tag{37}$$

where $x \in \mathbb{R}^6$ denotes a vector of position and orientation with coordinates in the earth-fixed frame, $\dot{x} \in \mathbb{R}^6$ denotes a vector of linear and angular velocities with coordinates in the earth-fixed frame, and $\nu \in \mathbb{R}^6$ denotes a vector of linear and angular velocities with coordinates in the body-fixed frame. The inertial effects, centripetal-Coriolis effects, hydrodynamic damping effects, gravitational effects, and control input in the earth-fixed frame can be represented by $\overline{M} : \mathbb{R}^6 \to \mathbb{R}^{6 \times 6}$, $\overline{C} :$

Table 3: Performance Comparison

| Architecture | | Adaptation Law | $\|e\|_{\mathrm{RMS}}$ (deg) | $\|u\|_{\mathrm{RMS}}$ (Nm) | Function error on-trajectory $(\mathrm{rad/s^2})$ | Final mean function error on test data $(\mathrm{rad/s^2})$ |
|---|---|---|---|---|---|---|
| Layers | Neurons | | | | | |
| 3 | 3 | Tracking Error-Based | 0.731 | 7.484 | 0.430 | 0.954 |
| | | Composite | 0.315 | 7.944 | 0.138 | 0.140 |
| | | % Decrease | 56.97 | -6.14 | 67.77 | 85.29 |
| 4 | 3 | Tracking Error-Based | 3.338 | 7.021 | 1.636 | 0.725 |
| | | Composite | 0.338 | 7.957 | 0.154 | 0.325 |
| | | % Decrease | 89.87 | -13.33 | 90.58 | 55.024 |
| 4 | 4 | Tracking Error-Based | 0.685 | 7.731 | 0.426 | 0.759 |
| | | Composite | 0.309 | 7.957 | 0.132 | 0.154 |
| | | % Decrease | 54.85 | -2.87 | 68.93 | 79.64 |
| 5 | 5 | Tracking Error-Based | 0.664 | 7.800 | 0.395 | 1.22 |
| | | Composite | 0.307 | 7.955 | 0.131 | 0.342 |
| | | % Decrease | 53.69 | -1.99 | 66.87 | 72.04 |
| 5 | 10 | Tracking Error-Based | 0.351 | 7.940 | 0.192 | 1.010 |
| | | Composite | 0.308 | 7.959 | 0.130 | 0.110 |
| | | % Decrease | 12.41 | -0.235 | 32.13 | 89.07 |
| 10 | 10 | Tracking Error-Based | 0.584 | 8.826 | 1.330 | 2.624 |
| | | Composite | 0.307 | 7.965 | 0.130 | 0.206 |
| | | % Decrease | 47.34 | 9.75 | 90.22 | 92.15 |

$\mathbb{R}^6 \times \mathbb{R}^6 \times \mathbb{R}^6 \to \mathbb{R}^{6\times6}$, $\overline{D}: \mathbb{R}^6 \times \mathbb{R}^6 \to \mathbb{R}^{6\times6}$, $\overline{G}: \mathbb{R}^6 \to \mathbb{R}^6$, and $\tau_n: \mathbb{R}_{\geq0} \to \mathbb{R}^6$, respectively. The velocities in the body-fixed frame can be related to the velocities in the earth-fixed frame using the relation

$$\dot{x} = J(x)\nu, \tag{38}$$

where $J: \mathbb{R}^6 \to \mathbb{R}^{6\times6}$ is a Jacobian transformation matrix relating the two frames (Fischer et al., 2014, Equation (2)). Thus, the dynamics in (37) can be represented using Eq. (1) from the manuscript with

$$f(x,\dot{x}) = -\overline{M}^{-1}(x)\left(\overline{C}(x,\dot{x},\nu)\dot{x} + \overline{D}(x,\nu)\dot{x} + \overline{G}(x)\right)$$

and

$$g(x,\dot{x}) = \overline{M}^{-1}(x).$$

Using the kinematic transformation in (38), the earth–fixed dynamics in (37) can be expressed using body-fixed dynamics as $\overline{M} = J^{-\top}MJ^{-1}$, $\overline{C} = J^{-\top}\left[C(\nu) - MJ^{-1}\dot{J}\right]J^{-1}$, $\overline{D} = J^{-\top}D(\nu)J^{-1}$, $\overline{G} = J^{-\top}G$, and $\tau_n = J^{-\top}\tau_b$, where $M \in \mathbb{R}^{6\times6}$, $C: \mathbb{R}^6 \to \mathbb{R}^{6\times6}$, $D: \mathbb{R}^6 \to \mathbb{R}^{6\times6}$, $G: \mathbb{R}^6 \to \mathbb{R}^6$, and $\tau_b: \mathbb{R}_{\geq0} \to \mathbb{R}^6$ denote the inertial effects, centripetal-Coriolis effects, hydrodynamic damping effects, gravitational effects, and control input in the body-fixed frame, respectively. The inertial effects, centripetal-Coriolis effects, and hydrodynamic damping effects in the body-fixed effects can be expressed as (Dixon, Behal, Dawson, and Nagarkatti, 2003, Equation (2.246))

$$M = \mathrm{diag}\{m_1, m_2, m_3, m_4, m_5, m_6\},$$
$$D = \mathrm{diag}\{d_{11} + d_{12}|\nu(1)|, d_{21} + d_{22}|\nu(2)|, d_{31} + d_{32}|\nu(3)|,$$
$$d_{41} + d_{42}|\nu(4)|, d_{51} + d_{52}|\nu(5)|, d_{61} + d_{62}|\nu(6)|\},$$

$$V_m = \begin{bmatrix} 0 & 0 & 0 & 0 & m_3\nu_3 & -m_2\nu_2 \\ 0 & 0 & 0 & -m_3\nu_3 & 0 & m_1\nu_1 \\ 0 & 0 & 0 & m_2\nu_2 & -m_1\nu_1 & 0 \\ 0 & m_3\nu_3 & -m_2\nu_2 & 0 & m_6\nu_6 & -m_5\nu_5 \\ -m_3\nu_3 & 0 & m_1\nu_1 & -m_6\nu_6 & 0 & m_4\nu_4 \\ m_2\nu_2 & -m_1\nu_1 & 0 & m_5\nu_5 & -m_4\nu_4 & 0 \end{bmatrix},$$

where the numerical values of mass, inertia, and damping parameters listed in Table 4 were used. The considered UUV is neutrally buoyant, thus $G = 0_{6\times1}$. The desired trajectory was selected as a

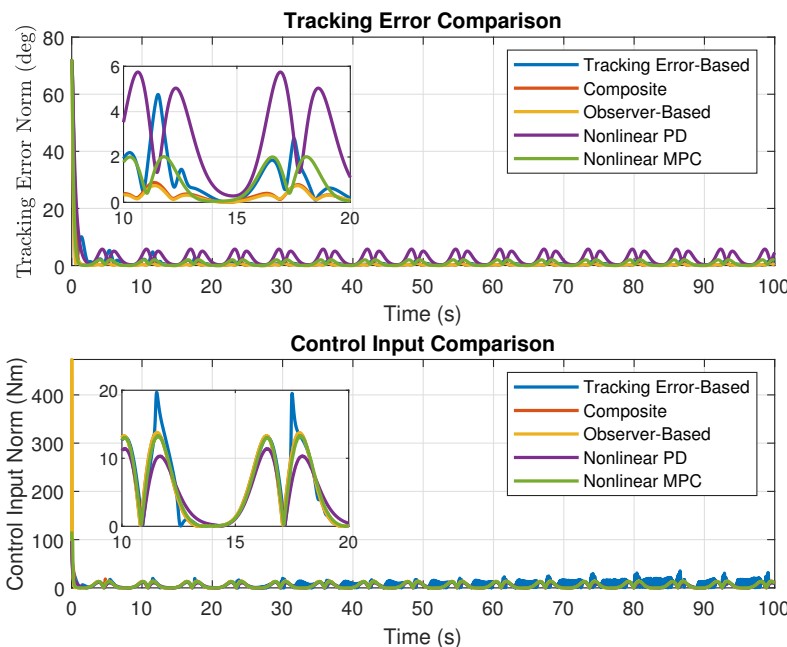

Figure 4: Comparative plots of the tracking error norm and control input norm along the trajectory with the developed and baseline controllers.

helical trajectory given by $x_d(t) = [2\cos(0.5t)\,\mathrm{m}, 2\sin(0.5t)\,\mathrm{m}, 0.1t\,\mathrm{m}, 0\,\mathrm{rad}, 0\,\mathrm{rad}, -0.125t\,\mathrm{rad}]^\top$, and the system was initialized with $x(0) = [-0.5\,\mathrm{m}, -0.5\,\mathrm{m}, -0.5\,\mathrm{m}, 0\,\mathrm{rad}, 0\,\mathrm{rad}, 0\,\mathrm{rad}]^\top$ and $\dot{x}(0) = [0_{1\times3}\,\mathrm{m/s}, 0_{1\times3}\,\mathrm{rad/s}]^\top$. The following gains were used in the simulation: $\alpha_1 = 5$, $\alpha_2 = 10$, $\alpha_3 = 40$, $k_r = 20$, $k_f = 20$, $k_{\hat{\theta}} = 0.0001$, $\Gamma(0) = 0.5I_{221}$, $\beta = 10$. The weights are randomly initialized from the distribution $U(-0.5, 0.5)$. Similar to the two-link manipulator, for a fair comparison, the set of gains common to the developed and baseline methods were selected to be exactly the same. The MPC was implemented in a similar manner as the two-link manipulator (see Appendix D.1.1 for details), except with optimizer with $Q_e = I$, $Q_r = I$, $R = 10I$, $N = 5$ steps, and bounded control input search space with upper and lower bounds of 5 N and -5 N, respectively, for every linear control input, and 5 Nm and -5 Nm, respectively, for every angular control input, as these values were empirically found to yield the most desirable performance.

Table 4: UUV System Parameters (Dixon et al., 2003, Equation (2.247))

| $m_1 = 215$ kg | $d_{11} = 70$ Nm·sec | $d_{41} = 30$ Nm·sec |
|---|---|---|
| $m_2 = 265$ kg | $d_{12} = 100$ N·sec$^2$ | $d_{42} = 50$ N·sec$^2$ |
| $m_3 = 265$ kg | $d_{21} = 100$ Nm·sec | $d_{51} = 50$ Nm·sec |
| $m_4 = 40$ kg·m$^2$ | $d_{22} = 200$ N·sec$^2$ | $d_{52} = 100$ N·sec$^2$ |
| $m_5 = 80$ kg·m$^2$ | $d_{31} = 200$ Nm·sec | $d_{61} = 50$ Nm·sec |
| $m_6 = 80$ kg·m$^2$ | $d_{32} = 50$ N·sec$^2$ | $d_{62} = 100$ N·sec$^2$. |

## D.3 BETTER RESOLUTION FIGURES

In the interest of plots in Figures 1-3 had to be shrinked. However, for a better visualization, the plots are reproduced in Figures 4-9.

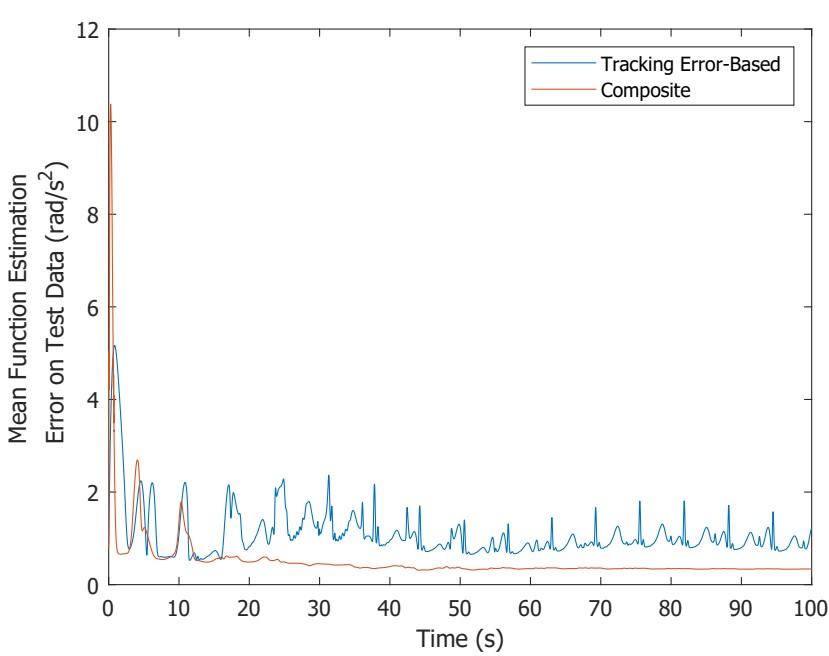

Figure 5: Comparative plots of the mean of function estimation error norm $\left\| f(x,\dot{x}) - \Phi\left(X,\hat{\theta}\right) \right\|$ on the test dataset.

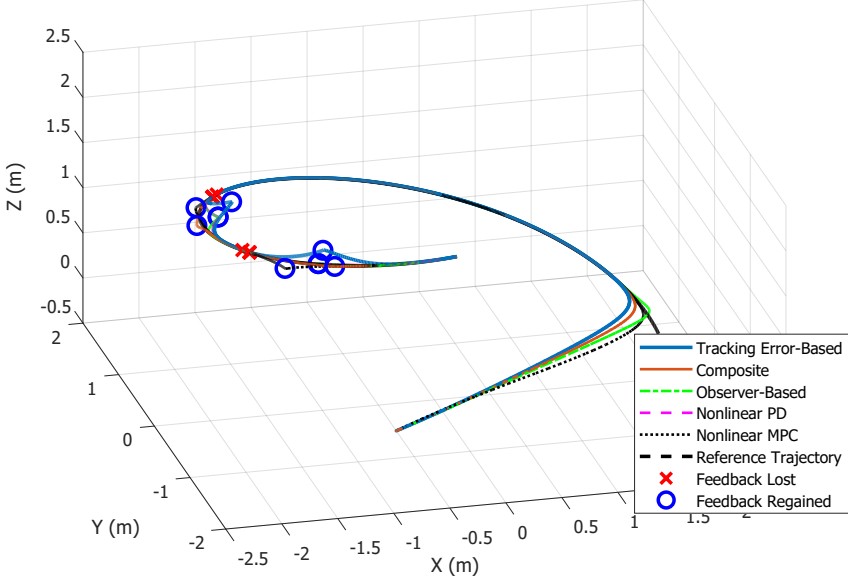

Figure 6: Plot of the position trajectories taken by the UUV using the developed and baseline methods. The plot is restricted to the trajectories from the time-interval [0, 9] seconds for visual clarity. The points where each controller loses feedback are marked by a red × and points where each controller regains feedback are marked by a blue ⃝.

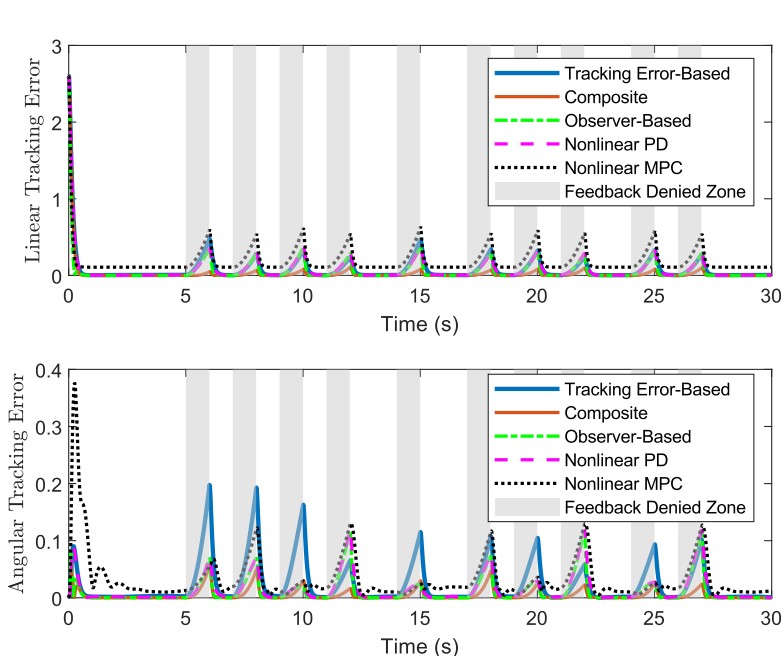

Figure 7: Comparative plots of the linear tracking error norm (m) and angular tracking error norm (rad) for the UUV. The time intervals corresponding to the feedback denied zones are marked in grey patches.

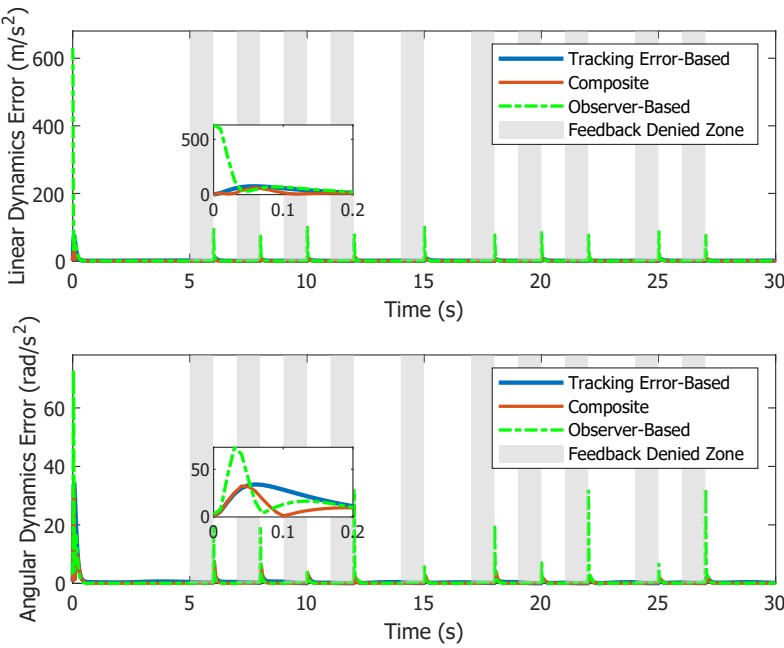

Figure 8: Comparative plots of the estimation error norm (i.e., $\left\| f(x, \dot{x}) - \Phi\left(X, \hat{\theta}\right) \right\|$ during feedback availability and $\left\| f(x, \dot{x}) - \Phi\left(X_d(t), \hat{\theta}\right) \right\|$ during loss of feedback) for the UUV with the developed and baseline methods.

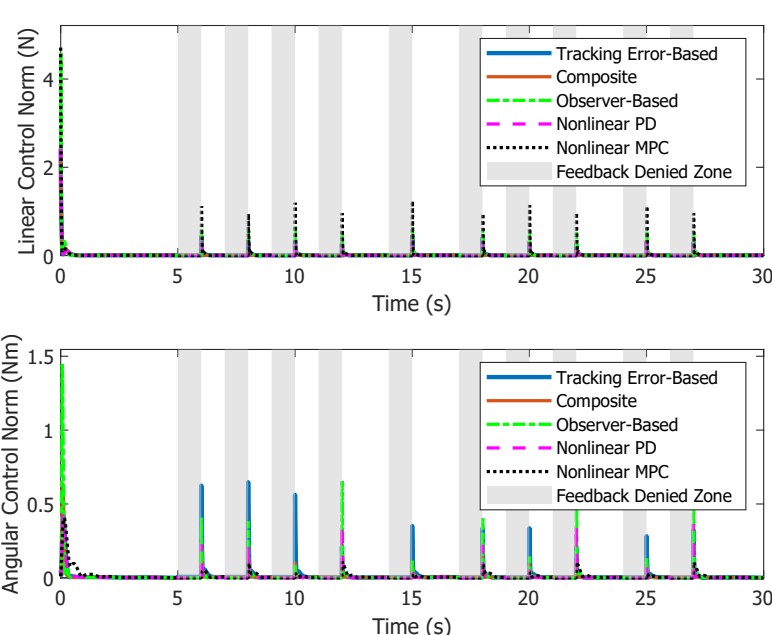

Figure 9: Comparative plots of the linear and angular control input norms for the UUV.

# E  DWELL-TIME CONDITIONS FOR STABLE OPERATION UNDER INTERMITTENT LOSS OF STATE FEEDBACK

Since the DNN identifies the system dynamics, the identified DNN estimates could be used to predict the uncertainty when the state feedback is intermittently lost. Let $i \in \mathbb{Z}_{\geq 0}$ denote the time index such that the state feedback is available in the time interval $[t_{2i}, t_{2i+1})$ and unavailable in the time interval $[t_{2i+1}, t_{2i+2})$ for all $i \in \mathbb{Z}_{\geq 0}$. During the time interval $[t_{2i}, t_{2i+1})$, when the feedback is available, the developed composite adaptive control and adaptation laws developed in the manuscript are used for all $i \in \mathbb{Z}_{\geq 0}$. However, during the time interval $[t_{2i+1}, t_{2i+2})$ when the state feedback is unavailable, the control input is designed to be an open-loop controller based on the last DNN weight estimate that was identified the feedback was available. The open-loop controller is given by

$$u = g^{+}(x_d(t), \dot{x}_d(t)) \left( \ddot{x}_d(t) - \Phi\left(X_d(t), \hat{\theta}(t_{2i+1})\right) \right). \tag{39}$$

Substituting (39) into $\ddot{x} = f(x, \dot{x}) + g(x, \dot{x})u$, subtracting $\ddot{x}_d$ on both sides, adding and subtracting $\Phi\left(X_d, \hat{\theta}(t_{2i+1})\right)$, and rearranging terms yields

$$\begin{aligned}
\ddot{e} = {} & \Phi(X, \theta^*) - \Phi\left(X_d(t), \hat{\theta}(t_{2i+1})\right) + \varepsilon(X) \\
& + \left(g(x, \dot{x})g^{+}(x_d(t), \dot{x}_d(t)) - I_n\right) \left(\ddot{x}_d(t) - \Phi\left(X_d(t), \hat{\theta}(t_{2i+1})\right)\right). 
\end{aligned} \tag{40}$$

For the purpose of this section, it is assumed the drift dynamics $f$ are globally Lipschitz with a Lipschitz constant $\varpi \in \mathbb{R}_{>0}$, and the control effectiveness and its pseudoinverse, $g$ and $g^{+}$, are globally bounded functions without bounds $\overline{g}, \overline{g^{+}}$ such that $\|g(x, \dot{x})\| \leq \overline{g}$ and $\|g^{+}(x, \dot{x})\| \leq \overline{g^{+}}$. The global Lipschitzness of $f$ is assumed in order to rule out the possibility of the drift dynamics causing finite-time escape during the absence of state-feedback. Such an assumption is reasonable since finite-time escape is usually not inherent to the uncontrolled dynamics for most practical systems of interest. Additionally, assuming that $g$ and $g^{+}$ are bounded is reasonable for most practical engineering systems, since the control effectiveness term usually results from the inertia matrix or the kinematic Jacobian of the system, and systems that may have potentially singular kinematic Jacobians in practice are not considered here. Additionally, in this section, a requirement is imposed on the selected

DNN $\Phi$ to contain bounded globally Lipschitz activation functions. Thus, using bounds on $g$, $g^+$, $\Phi\left(X_d, \hat{\theta}(t_{2i+1})\right)$, $\tilde{\theta}(t_{2i+1})$, and $\ddot{x}_d$, it can be shown that there exists constants $L_U, \delta_U \in \mathbb{R}_{>0}$ such that $\|\ddot{e}\| \leq L_U \|e\| + L_U \|\dot{e}\| + \delta_U$. Using the relations $r = \dot{e} + \alpha_1 e$ and $\ddot{e} = \dot{r} - \alpha_1 \dot{e}$ yields the inequality $\|\dot{r}\| \leq \left(\alpha_1^2 + L_U \alpha_1 + L_U\right) \|e\| + \left(L_U + \alpha_1\right) \|r\| + \delta_U$. Additionally, since $f$ is considered to be globally Lipschitz in this section, it follows that $\left\|\frac{\partial f}{\partial x}\right\|, \left\|\frac{\partial f}{\partial \dot{x}}\right\| \leq \varpi$. As a result, it can be shown that $\left\|\dot{f}\right\| \leq \left(2\alpha_1^2 + (L_U + 1)\alpha_1 + L_U\right)\varpi \|e\| + \left(L_U + 2\alpha_1 + 1\right)\varpi \|r\| + \left(\delta_U + \overline{\dot{x}_d} + \overline{\ddot{x}_d}\right)\varpi$. During the loss of state feedback, all observer and adaptive update laws are selected to be zero, i.e., $\dot{\hat{r}} = 0$, $\dot{\hat{f}} = 0$, and $\dot{\hat{\theta}} = 0$.

The growth of the Lyapunov function in (18) is examined using the bounds on $\dot{r}$ and $\dot{f}$ to analyze the growth of the error states during the loss of state feedback. By successive use of Holder's and Young's inequalities, it can be shown that $\dot{V} \leq \lambda_U V + \Delta_U$, when feedback is unavailable, where $\lambda_U \triangleq 2\max(\frac{3\alpha_1^2 + L_U \alpha_1 + L_U + 1}{2} + (2\alpha_1^2 + (L_U + 1)\alpha_1 + L_U)^2\varpi^2 + (\alpha_1^2 + L_U\alpha_1 + L_U)^2, \frac{\alpha_1^2 + (L_U + 2)\alpha_1 + 3L_U + \delta_U + 1}{2} + (L_U + 2\alpha_1 + 1)^2\varpi^2 + (L_U + \alpha_1)^2, \frac{1}{2})$ and $\Delta_U \triangleq \frac{\delta_U}{2} + \delta_U^2 + (\delta_U + \overline{\dot{x}_d} + \overline{\ddot{x}_d})^2\varpi^2$. Solving for $V$ for yields $V(t) \leq V(t_{2i+1})e^{\lambda_U(t - t_{2i+1})} + \frac{\Delta_U}{\lambda_U}(e^{\lambda_U(t - t_{2i+1})} - 1)$ for all $(t, i) \in [t_{2i+1}, t_{2i+2}) \times \mathbb{Z}_{\geq 0}$. Then applying the bounds in (19) and taking the square root yields

$$\|z(t)\| \leq \sqrt{\frac{\lambda_2}{\lambda_1} \|z(t_{2i+1})\|^2 e^{\lambda_U(t - t_{2i+1})} + \frac{2\Delta_U}{\lambda_1 \lambda_U}\left(e^{\lambda_U(t - t_{2i+1})} - 1\right)}, \tag{41}$$

for all $(t, i) \in [t_{2i+1}, t_{2i+2}) \times \mathbb{Z}_{\geq 0}$.

When the system regains feedback, the condition $z(t_{2i+2}) \in \mathcal{S}$ needs to be satisfied for the composite adaptive Lb-DNN to yield the results in Theorem 1 of the manuscript. Imposing this condition yields the following condition for maximum dwell time during which feedback can be unavailable without affecting the UUB properties of the resulting switched system,

$$(t_{2i+2} - t_{2i+1}) \leq \frac{1}{\lambda_U} \ln\left(\frac{\frac{\lambda_1}{\lambda_2}\chi^2 + \frac{2\Delta_U}{\lambda_1 \lambda_U} - \frac{c}{\lambda_3}}{\frac{\lambda_2}{\lambda_1}\|z(t_{2i+1})\|^2 + \frac{2\Delta_U}{\lambda_1 \lambda_U}}\right), \tag{42}$$

for $(t, i) \in [t_{2i+1}, t_{2i+2}) \times \mathbb{Z}_{\geq 0}$. The maximum dwell time in (42) should be positive for the system to sufficiently allow feedback unavailability, which holds when $\frac{\lambda_1}{\lambda_2}\chi^2 + \frac{2\Delta_U}{\lambda_1 \lambda_U} - \frac{c}{\lambda_3} > \frac{\lambda_2}{\lambda_1}\|z(t_{2i+1})\|^2 + \frac{2\Delta_U}{\lambda_1 \lambda_U}$. Imposing this condition on $\|z(t_{2i+1})\|^2$ and using Theorem 1 of the manuscript yields the following condition for minimum dwell time during which the feedback should be available

$$(t_{2i+1} - t_{2i}) \geq \frac{\lambda_2}{\lambda_3} \ln\left(\frac{\frac{\lambda_2}{\lambda_1}\|z(t_{2i})\|^2}{\frac{\lambda_1^2}{\lambda_2^2}\chi^2 - \frac{\lambda_2 c}{\lambda_1 \lambda_3} - \frac{\lambda_1 c}{\lambda_2 \lambda_3}}\right), \tag{43}$$

for all $(t, i) \in [t_{2i+1}, t_{2i+2}) \times \mathbb{Z}_{\geq 0}$. Note that it is permissible to obtain negative values for the dwell-time in (43), since a negative minimum dwell-time for feedback availability would imply the stability guarantees hold even if feedback continues to be unavailable after the time instance $t_{2i}$. Additionally, the size of set $\mathcal{D}$, i.e., $\chi$ needs to be selected according to $\chi > \sqrt{\frac{\lambda_2^3 c}{\lambda_1^3 \lambda_3} + \frac{\lambda_2 c}{\lambda_1 \lambda_3}}$, to ensure a positive denominator in (43), thus guaranteeing the feasibility of the minimum dwell-time condition.

# F   EXTENSION TO UNDERACTUATED SYSTEMS: NONHOLONOMIC MOBILE ROBOT

Although the main control development in Section 2 is dedicated to fully-actuated systems, the developed method can be extended to under-actuated systems on a case-by-case basis by a specialized treatment. A brief illustration of the extension to nonholonomic mobile robots is provided in this section, based on the previous work Fierro and Lewis (1998), where a shallow NN-based controller for fully-actuated systems in Lewis et al. (1996b) was extended for nonholonomic mobile robots.

A mobile robot having an $n$-dimensional configuration space with generalized coordinates $q \in \mathbb{R}^n$ subjected to $m$ constraints can be described by

$$M(q)\ddot{q} + V_m(q,\dot{q})\dot{q} + F(\dot{q}) = B(q)\tau - A^\top(q)\lambda, \tag{44}$$

where $M(q) \in \mathbb{R}^{n \times n}$ is a symmetric, positive-definite inertia matrix, $V_m(q,\dot{q}) \in \mathbb{R}^{n \times n}$ is the centripetal-Coriolis matrix, $F(\dot{q}) \in \mathbb{R}^n$ denotes the surface friction, $B(q) \in \mathbb{R}^{n \times (n-m)}$ denotes the input transformation matrix, $A(q) \in \mathbb{R}^{m \times n}$ denotes the matrix associated with constraints, and $\lambda \in \mathbb{R}^m$ denotes the constraint forces. The kinematic equality constraints can be expressed as

$$A(q)\dot{q} = 0. \tag{45}$$

Let $S(q) \in \mathbb{R}^{n \times (n-m)}$ be a matrix of rank $(n-m)$ satisfying

$$S^\top(q)A^\top(q) = 0. \tag{46}$$

For more details on the structures of the above matrices, the reader is referred to Fierro & Lewis (1998).

Due to (45) and (46), there exists an auxiliary vector time function $v(t) \in \mathbb{R}^{n-m}$ such that, for all $t$,

$$\dot{q} = S(q)v(t). \tag{47}$$

Therefore, the system in (44) can be described using a transformed representation given by

$$\begin{aligned} \dot{q} &= S(q)v \\ \overline{M}(q)\dot{v} + \overline{V_m}(q,\dot{q})v + F(\dot{q}) &= \overline{B}(q)\tau \end{aligned} \tag{48}$$

where $\overline{M}(q) \triangleq S^\top(q)M(q)S(q)$ and $\overline{V_m}(q,\dot{q}) \triangleq S^\top(q)\left(M(q)\dot{S}(q) + V_m(q,\dot{q})S(q)\right)$ denote the transformed inertia and centripetal Coriolis matrices, and $\overline{B}(q) \triangleq S^\top(q)B(q)$ denotes a transformed input matrix which is invertible. For a wheeled mobile robot, $q = [x,y,\vartheta]^\top$, where $x,y$ denotes its position coordinates, and $\vartheta$ denotes its orientation. To formulate a feasible trajectory tracking problem, let the reference cart be defined as

$$\begin{aligned} \dot{x}_r &= \mathrm{v}_r \cos\vartheta_r \\ \dot{y}_r &= \mathrm{v}_r \sin\vartheta_r \\ \dot{\theta}_r &= w_r \\ q_r &= [x_r, y_r, \vartheta_r]^\top, \end{aligned} \tag{49}$$

where $\mathrm{v}_r \in \mathbb{R}_{>0}$ denotes the reference speed, $w_r \in \mathbb{R}$ denotes the reference angular velocity, and $q_r$ denotes the reference trajectory generated by the cart. The tracking error is expressed in the basis of a frame linked to the mobile robot as

$$\begin{bmatrix} e_1 \\ e_2 \\ e_3 \end{bmatrix} = \begin{bmatrix} \cos\vartheta & \sin\vartheta & 0 \\ -\sin\vartheta & \cos\vartheta & 0 \\ 0 & 0 & 1 \end{bmatrix} \begin{bmatrix} x_r - x \\ y_r - y \\ \vartheta_r - \vartheta \end{bmatrix}. \tag{50}$$

An auxiliary velocity term that can achieve tracking, when only the kinematics is considered, is given by

$$v_c = \begin{bmatrix} \mathrm{v}_r \cos e_3 + K_1 e_1 \\ w_r + K_2 \mathrm{v}_r e_2 + K_3 \mathrm{v}_r \sin e_3 \end{bmatrix}, \tag{51}$$

where $K_1, K_2, K_3 \in \mathbb{R}_{>0}$ are constant control gains. Let the error in tracking the auxiliary velocity $v_c$ be defined as

$$e_c = v_c - v. \tag{52}$$

Differentiating on both sides of (52), multiplying by $\overline{M}(q)$ on both sides, and substituting (48) yields

$$\overline{M}(q)\dot{e}_c = \overline{M}(q)\dot{v}_c + \overline{V_m}(q,\dot{q})v + F(\dot{q}) - \overline{B}(q)\tau. \tag{53}$$

By pre-multiplying the both sides of (53) by $\overline{M}^{-1}(q)$, the following form of the system is obtained given by

$$\dot{e}_c = F(q, \dot{q}, e_c, \mathrm{v}_r, w_r, \dot{\mathrm{v}}_r, \dot{w}_r) + G(q)\tau, \tag{54}$$

where $X_m \triangleq \left[ q^\top, \dot{q}^\top, e_c^\top, \mathrm{v}_r^\top, w_r^\top, \dot{\mathrm{v}}_r^\top, \dot{w}_r^\top \right]^\top$ is a concatenated vector, $F(X_m) \triangleq \dot{\mathrm{v}}_c + \overline{M}^{-1}(q) \left( \overline{V_m}(q, \dot{q}) \mathrm{v} + F(\dot{q}) \right)$ is the uncertainty in the system, and $G(q) \triangleq -\overline{M}^{-1}(q) \overline{B}(q)$ denotes the known control effectiveness. Using the universal function approximation property, the term $F(X_m)$ can be approximated as

$$F(X_m) = \Phi(X_m, \theta^*) + \varepsilon(X_m),$$

for all $X_m \in \Omega$, where $\Omega \subset \mathbb{R}^{\dim(X_m)}$ is a compact set, and $\sup_{X_m \in \Omega} \|\varepsilon(X_m)\| \leq \bar{\varepsilon}$ for a prescribed accuracy $\bar{\varepsilon} \in \mathbb{R}_{>0}$. Therefore, the error system in (54) can be rewritten as

$$\dot{e}_c = \Phi(X_m, \theta^*) + \varepsilon(X_m) + G(q)\tau.$$

The control input is designed as

$$\tau = G^{-1}(q) \left( -\Phi\left(X, \hat{\theta}\right) - K_4 e_c \right). \tag{55}$$

To formulate the prediction error for constructing the composite adaptive update, a dynamic state-derivative estimator can be designed as

$$
\begin{aligned}
\dot{\hat{e}}_c &= \hat{F} + G(q)\tau + K_5 \tilde{e}_c \\
\dot{\hat{F}} &= K_6 \left( \dot{\hat{e}}_c + K_4 \tilde{e}_c \right) + \tilde{e}_c,
\end{aligned} \tag{56}
$$

and the corresponding observer error system is given by

$$
\begin{aligned}
\dot{\tilde{e}}_c &= \widetilde{F} - K_5 \tilde{e}_c \\
\dot{\widetilde{F}} &= \dot{F} - K_6 \widetilde{F} + \tilde{e}_c,
\end{aligned}
$$

where $\tilde{e}_c \triangleq e_c - \hat{e}_c$ and $\widetilde{F} = F(X_m) - \hat{F}$. Accordingly, the prediction error is formulated as

$$E = \hat{F} - \Phi\left(X, \hat{\theta}\right). \tag{57}$$

Using the prediction error $E$ and velocity error $e_c$, the adaptive update law for $\hat{\theta}$ is designed as

$$\dot{\hat{\theta}} = \mathrm{proj}\left( -k_{\hat{\theta}} \Gamma(t)\hat{\theta} + \Gamma(t)\Phi'^\top\left(X, \hat{\theta}\right)(e_c + \alpha_3 E) \right).$$

To show stability guarantees, the Lyapunov-based stability analysis for this system can be performed in a similar manner as Theorem 1, using the candidate Lyapunov function

$$
\begin{aligned}
V_M &= \frac{1}{2} e_1^\top e_1 + \frac{1}{2} e_2^\top e_2 + K_3 \mathrm{v}_r (1 - \cos e_3) + \frac{1}{2} e_c^\top e_c \\
&\quad + \frac{1}{2} \tilde{e}_c^\top \tilde{e}_c + \frac{1}{2} \widetilde{F}^\top \widetilde{F} + \frac{1}{2} \tilde{\theta}^\top \Gamma^{-1} \tilde{\theta}.
\end{aligned}
$$

For more details on the time-derivative calculations of the first four terms in $V_M$, the reader is referred to Fierro & Lewis (1998).

