# OpenReview forum: "Simultaneous Online System Identification and Control using Composite Adaptive Lyapunov-Based Deep Neural Networks"
_ICLR.cc/2025/Conference — Submitted to ICLR 2025_

### Official Review · Reviewer_N4X4 · 2024-10-23

**Soundness:** 2
**Presentation:** 2
**Contribution:** 3
**Rating:** 8
**Confidence:** 4

**Summary:**

This paper presents a kind of Lpapunov-based adaptive framework, that can update all layers of DNN. The proposed method can handle nonlinear-in-parameters uncertainties. Moreover, a dynamic state-derivative estimator is utilized to obtain the state-derivative information. Overall, some novel theoretical results are developed in this paper with rigid proofs. The presentation is also clear. However, some drawbacks exist and many improvements can be further considered. There are some inappropriate statements and comparisons. The simulation tests are not enough to show its efficiency. Please refer to below for more details.

**Strengths:**

1. The main contribution is that a Lpapunov-based adaptive framework is proposed to update **all layers** of DNN.
2. Rigid convergence analysis.
3. Two applied examples, despite only numerical simulations.

**Weaknesses:**

1. The Abstract is too long, preventing the reader from capturing key points quickly. It is recommended to only highlight the main contributions in the Abstract and technique details can be removed.

2. It is claimed that the tracking, state-derivative, and weight estimation errors can be guaranteed to converge to bounded sets. **The factors that determine the upper bounds** of convergence sets should be provided in the Abstract.

3. The previous work (OConnell, 2022) is compared in the Intro. It is claimed the limitation of the composite adaptive approach used by OConnell, 2022, is the inner-layer weight cannot be online updated. However, the considered case of OConnell, 2022 is different from the one of this paper. OConnell mainly focuses on a composited disturbance, which comes from external disturbance and internal state-related uncertainties. The last layer of DNN is updated online to handle external disturbances, while the inner layers correspond to internal state-related uncertainties, which would not change in application. However, the internal state-related uncertainty is mainly considered in this paper, i.e., $f({x, \dot{x}})$. **The direct comparison with (OConnell, 2022) is inappropriate**.

4. One important problem is only simulation examples are demonstrated in this paper, and no noises exist in the measured states, despite the theorems that seem to be relatively complete. A small upper bound of the convergence set depends on a large gain. However, the gain may enlarge the noises in a real system. Thus, **the effect of the real application is questionable**.

5. In the simulation of two link manipulators, it is recommended to cover the ESO comparison and the composite adaptive method developed in (Slotine and Li, 1989). The gain selection strategy for all comparison methods should be provided to ensure fairness.

**Questions:**

1. If the proposed framework could be combined with offline learning?  It seems the proposed update strategy only relies on current measurements and has no historical data-mining procedure. It is a kind of traditional adaptive control, instead of modern data-based learning. Maybe a control journal is more applicable to this paper.

2. If the considered uncertainty $f({x, \dot{x}})$ can be extended to the composited disturbance, like $f({x, \dot{x}, d})$, where $d$ denotes the external disturbance. It will be valuable in real applications.

---

> ### Author Response · Authors · 2024-11-23
>
> We thank the reviewer for the appreciation of our contributions, theoretical results, and presentation. Based on the reviewer's suggestion we removed the technical details in the abstract to make it more concise. Additionally, we have added the following sentence to the abstract: “under the persistence of excitation (PE) condition, the tracking and weight estimation errors are shown to exponentially converge to a neighborhood of the origin, where the rate of convergence and the size of this neighborhood depends on the gains and a factor quantifying PE”.
>
> **Regarding composited disturbances**
>
> We apologize for the brief statement about the previous work (O'Connell, 2022) in the introduction which may have caused confusion. To make the discussion about (OConnell et al., 2022) more precise, we have moved the discussion about this reference to the Related Work Section in Appendix B.2. For the reviewer's convenience, we are reproducing the discussion here:
>
> “The recent result in (O'Connell, Shi, Shi, Azizzadenesheli, Anandkumar, Yue, and Chung, 2022) developed a new learning representation uncertainties involving a composited disturbance given by $f(x,\dot{x},w)=\phi\left(x,\dot{x}\right)a\left(w\right)$, where $\phi\left(\cdot\right)$ denotes a basis function that is learned using a DNN and $a\left(w\right)$ denotes a set of linear parameters accounting for an unknown disturbance time-varying disturbance $w$. Since $f$ is linear in terms of $a$, the composite adaptive approach from (Slotine & Li, 1989) is used to design an adaptation law $\dot{\hat{a}}$ to update the estimates of a given by $\hat{a}$. To obtain a disturbance-invariant representation of $\phi$ using DNNs, a domain adversarially invariant meta-learning (DAIML) algorithm is developed to train the DNN offline. To the best of our knowledge, this is the only existing work using a composite adaptive approach in the context of deep learning-based control. However, since the DNN learning $\phi\left(x,\dot{x}\right)$ has an NIP structure, the aforementioned challenges apply for constructing a Lyapunov-based online adaptation law.”
>
> Please note **this is not a criticism of (O'Connell, 2021)**; we acknowledge this method involves a fundamentally different paradigm than ours. We strongly appreciate the DAIML approach to obtain a disturbance-invariant representation in (O'Connell, 2021). Since (O'Connell, 2021) is the only existing work using a composite adaptive approach in the context of deep learning-based control, we believe it deserves to be cited and discussed in the related works section.
>
> Furthermore, note that our developed method can be easily extended for composited disturbances. Note that the developed method is agnostic of the DNN architecture used. Therefore, for a composited disturbance $f(x,\dot{x},w)$, the representation $f(x,\dot{x},w)=\phi\left(x,\dot{x}\right)a\left(w\right)$ from (O'Connell, 2022) can be used. The term $\phi$ can be approximated as $\phi\left(x,\dot{x}\right)=\Phi\_\{1}\left(X,\theta\_{1}^*\right)+\varepsilon\_\{1}\left(X\right)$, where $\Phi\_\{1}\left(X,\theta\_{1}^*\right)$ is a DNN with an appropriate ideal parameter $\theta_1^*$, which yields $f(x,\dot{x},w)=\Phi\_1\left(X,\theta\_1^*\right)a\left(w\right)+\varepsilon\_1\left(X\right)a\left(w\right)$. Then $\theta^{\*\top}=[a(w)^\top \theta\_1^{\*\top}]^{\top}$ can be constructed as an augmented parameter to define the parameterization $\Phi\left(X,\theta^*\right)=\Phi\_{1}\left(X,\theta\_{1}^*\right)a\left(w\right)$ and $\varepsilon\left(X\right)=\varepsilon\_1\left(X\right)a\left(w\right)$, which yields $f(x,\dot{x},w)=\Phi\left(X,\theta^*\right)+\varepsilon\left(X\right)$. In this case, $\theta^*$ becomes a time-varying parameter. The analysis in (O'Connell, 2022) treated $a(w)$ as a bounded time-varying parameter with a bounded time-derivative. This argument can then be used to obtain that $\theta^*$ and $\dot{\theta}^*$ are bounded. In the Lyapunov-based analysis the extra term $\tilde{\theta}^{\top}\dot{\theta}^*$ would appear since $\theta^*$ is time-varying. The bound on $\dot{\theta}^*$ can be used to conclude exponential convergence of all the error states to the neighborhood of the origin in a similar manner as in the proof of Theorem 1, except that the size of this neighborhood now also depends on the bound on $\dot{\theta}^*$.

---

> ### Author Response · Authors · 2024-11-23
>
> **Revisions in Simulations**
>
> Based on the reviewer's suggestion of adding measurement noise to make the simulations more realistic, we performed all of the simulations again with measurement noise, and the results in the simulation section are now updated accordingly. Additionally, although the reviewer is correct that high gains can amplify measurement noise, this is true for all feedback controllers in general; the tradeoff between tracking performance, control effort, and noise sensitivity is well-known. Notably, instead of using numerical derivatives in our update laws, we use a dynamic state-derivative estimator which is shown to be robust to noise in Appendix A.2. As shown in the proof, the ultimate bound on state-derivative estimation error grows linearly with the bound on the noise $\bar{\delta}$.
>
> Furthermore, based on the reviewer's suggestion, we have also included the state-derivative observer-based disturbance rejection controller as a baseline for the two link manipulator. The gain selection strategy for ensuring fairness is described in Appendix D. For a fair comparison, the set of gains common to the developed and baseline methods were selected to be exactly the same. We did not perform a comparison with the composite adaptive method in (Slotine and Li, 1989) because the comparison would not be fair in our opinion. The method in (Slotine and Li, 1989) has an unfair advantage because it uses regressor information based on the structural knowledge of the manipulator dynamics. The DNN being a black box model does not have access to this information, so Slotine and Li's method is expected to perform better. This is for similar reasons as why a feedback linearization controller using the exact knowledge of $f(x,\dot{x})$ would be expected to perform better than Slotine and Li's method or our method.
>
> **Response to other queries**
>
> The proposed framework can be easily combined with offline learning by initializing the DNN weight estimates with pre-trained weights. In fact, initializing with pre-trained weights (e.g., especially using meta-learning approaches such as (O'Connell, 2021)) would likely further improve the controller performance. However, to better showcase the online learning performance with the composite adaptation law, we initialized the DNN with completely random weights in our simulations.
>
> Although it might appear the update strategy involves only the current state measurements, the least-squares update law does actually involve a historical data-mining procedure. Specifically, notice the least squares adaptation gain matrix $\Gamma(t)$ evolves according to Eq. (12) in the manuscript, which reads $\frac{d}{dt}\Gamma^{-1}(t)=-\beta(t)\Gamma^{-1}(t)+\Phi^{\prime\top}\left(X,\hat{\theta}\right)\Phi^{\prime}\left(X,\hat{\theta}\right)$. This equation reveals that the adaptation gain matrix $\Gamma(t)$ implicitly incorporates information from all past data points through the dynamics of $\Gamma^{-1}(t)$, which accumulate contributions from the regressor matrix $\Phi^{\prime\top}\left(X,\hat{\theta}\right)\Phi^{\prime}\left(X,\hat{\theta}\right)$ over time. As a result, the update law inherently processes historical data by integrating its influence into the adaptation dynamics, thereby enabling the least-squares framework to adapt based on both current and prior system behaviors.
>
> Regarding the comment “It is a kind of traditional adaptive control, instead of modern data-based learning. Maybe a control journal is more applicable to this paper”, we would like to point that our work lies at the intersection of adaptive control and deep learning. Specifically, the use of deep neural networks makes our work different from traditional adaptive control. Furthermore, ICLR explicitly invites contributions involving applications to robotics, autonomy, planning (which is the primary area of this submission).
>
> References:
>
> O’Connell, M., Shi, G., Shi, X., Azizzadenesheli, K., Anandkumar, A., Yue, Y. and Chung, S.J., 2022. Neural-fly enables rapid learning for agile flight in strong winds. Science Robotics, 7(66), p.eabm6597.
>
> Slotine, J.J.E. and Li, W., 1989. Composite adaptive control of robot manipulators. Automatica, 25(4), pp.509-519.

---

> > ### Comment · Reviewer_N4X4 · 2024-11-28
> > **Thanks for the reply**
> >
> > Thank you for the authors' responses！ Most of my comments have been addressed. I would like to discuss this 'control or learning' point further with the authors. The authors claim that the presented scheme represents an intersection of adaptive control and deep learning. However, one key advantage of deep learning is its ability to extract valuable insights from lots of historical data. In contrast, the proposed adaptive strategy relies solely on the current feedback state to update all layers. This suggests that the learned neural network is only applicable to the present moment, which raises questions about its alignment with deep learning principles, aside from the network structure as its adaptive regressor/basis. It's curious what the author would think about this.
> >
> > I acknowledge the valuable theoretical contribution, offering significant value from an adaptation perspective. I believe that conducting future real-world experiments will prove its validity and greatly enhance its contributions. I keep the score.

---

> ### Author Response · Authors · 2024-11-28
> **On the intersection of deep learning and adaptive control (Part 1)**
>
> We thank the reviewer for the constructive feedback and appreciation of our work. We wholeheartedly agree that conducting future real-world experiments will prove its validity and greatly enhance its contributions, and we plan on doing so.
>
> The reviewer has raised an intriguing question about the alignment of deep learning principles with our adaptive design. We would like to take this opportunity to explain the existing connections between the core principles of deep learning and adaptive control. The core ideas in deep learning and adaptive control are both rooted in the model-based regression problem. Traditional deep learning typically involves optimization methods such as stochastic gradient descent or ADAM to minimize the loss function offline over prior datasets to perform the regression. Least squares adaptive control essentially uses online gradient descent to minimize the least squares loss over the data encountered online so far by the system. For more details, the reviewer is referred to the recent work (Gaudio et. al., 2019) which demonstrated the equivalence between the machine learning and adaptive control paradigms.
>
> In a sense, the least squares adaptive approach is a recursive/continuous-time implementation of batch least squares update. Although it might appear from the adaptive update law that it is solely relying on the current state information, this appearance is misleading, especially in the case of least squares update. The historical data encountered by the system shows up implicitly in the weight estimates due to the previous state values, and the previously encountered data is not forgotten. Please see Part 2 of our response for an explicit mathematical illustration.
>
> The key takeaway is, adaptive control methods address the same underlying model-based regression/parameter estimation problem. Due to guarantees on accurate parameter estimation under the PE condition, the obtained model can generalize well-beyond the current or even off-trajectory datapoints. This is why in our simulations on the two-link, our developed method was able to achieve a good model generalization on the test dataset involving off-trajectory datapoints. Similarly, in the case of the UUV, we showed that the identified DNN model can be used open-loop to make predictions when the state feedback is lost. The open-loop controller using the DNN identified with our method was effective despite the loss of feedback due to its ability to generalize on newer data.
>
> Note that, in a sense, the PE condition is indicative of exploration or the richness of data encountered by the system online. Similar to how traditional deep learning methods would extract valuable information from the historical data, our method leverages PE condition to achieve accurate parameter estimation, thus enabling system identification and model's ability to generalize beyond the encountered trajectory. Furthermore, as stated before, the existing historical data can still be used to initialize the DNN by offline training, and the discussed meta learning based approaches are valuable in this regard. We believe such an approach might reduce the burden of online exploration and would be an interesting research direction for future work. The void that this work fills is the continued learning after task execution. Over the lifetime of the task execution, all the data is continuously being embedded within the DNN from the actual operating environment and operating conditions.
>
> References:
>
> Gaudio, J.E., Gibson, T.E., Annaswamy, A.M., Bolender, M.A. and Lavretsky, E., 2019, December. Connections between adaptive control and optimization in machine learning. In 2019 IEEE 58th Conference on Decision and Control (CDC) (pp. 4563-4568). IEEE.

---

> ### Author Response · Authors · 2024-11-28
> **On the intersection of deep learning and adaptive control (Part 2)**
>
> For ease of illustration, consider a simple neural network (NN) with a scalar input $x$, one hidden layer, and two scalar weights, given by:
>
> $\Phi(x, \hat{\theta}) = \hat{w}\_{1} \phi(\hat{w}\_{0} x),$
>
> with $\hat{\theta} = [\hat{w}\_{0}, \hat{w}\_{1}]^{\top}$ and Jacobian:
>
> $\Phi^{\prime}(x, \hat{\theta}) = \left[ \hat{w}\_{1} \phi^{\prime}(\hat{w}\_{0} x) x,\, \phi(\hat{w}\_{0} x) \right]^{\top}.$
>
> Furthermore, assume perfect tracking (zero tracking error), exact state-derivative estimates (i.e., $\hat{f} = f(x)$, exact ground truth information), let the forgetting factor $\beta = 0$, and omit the projection operator. In this case, the least squares adaptive update is given by:
>
> $\dot{\hat{\theta}} = -k_{\hat{\theta}} \Gamma(t) \hat{\theta} + \Gamma(t) \Phi^{\prime\top}(x, \hat{\theta}) \left( f(x) - \Phi(x, \hat{\theta}) \right)$
>
> $\dot{\Gamma} = -\Gamma \Phi^{\prime\top}(X, \hat{\theta}) \Phi^{\prime}(X, \hat{\theta}) \Gamma.$
>
> For Euler discretization, let the time step be $\Delta t$ and denote the discrete-time values at time step $n$ by $\hat{\theta}[n]$ and $\Gamma[n]$. Suppose the initial values are $\hat{\theta}[0] = \hat{\theta}\_{0}$ and $\Gamma[0] = \Gamma\_{0}$.
>
> ### Step 1 ($n = 1$)
>
> $\hat{\theta}[1] = \hat{\theta}[0] + \Delta t \left( -k_{\hat{\theta}} \Gamma[0] \hat{\theta}[0] + \Gamma[0] \Phi^{\prime\top}(x[0], \hat{\theta}[0]) \left( f(x[0]) - \Phi(x[0], \hat{\theta}[0]) \right) \right)$
>
> $\Gamma[1] = \Gamma[0] - \Delta t \Gamma[0] \Phi^{\prime\top}(X[0], \hat{\theta}[0]) \Phi^{\prime}(X[0], \hat{\theta}[0]) \Gamma[0]$
>
> ### Step 2 ($n = 2$)
>
> $\hat{\theta}[2] = \hat{\theta}[1] + \Delta t \left( -k_{\hat{\theta}} \Gamma[1] \hat{\theta}[1] + \Gamma[1] \Phi^{\prime\top}(x[1], \hat{\theta}[1]) \left( f(x[1]) - \Phi(x[1], \hat{\theta}[1]) \right) \right)$
>
> $\Gamma[2] = \Gamma[1] - \Delta t \Gamma[1] \Phi^{\prime\top}(X[1], \hat{\theta}[1]) \Phi^{\prime}(X[1], \hat{\theta}[1]) \Gamma[1]$
>
> ### Observing Dependencies
>
> - $\hat{\theta}[1]$ explicitly depends on $\hat{\theta}[0]$, $\Gamma[0]$, and the data $x[0]$ and $f(x[0])$.
> - $\hat{\theta}[2]$ explicitly depends on $\hat{\theta}[1]$, $\Gamma[1]$, and the data $x[1]$ and $f(x[1])$.
>
> Substituting $\hat{\theta}[1]$ into $\hat{\theta}[2]$ reveals that $\hat{\theta}[2]$ also indirectly depends on $\hat{\theta}[0]$, $\Gamma[0]$, and the old data $x[0]$ and $f(x[0])$.
>
> Thus, the updates are recursive and inherently carry information from previous states and data. This recursion ensures that historical data is not forgotten, aligning with the principle of online learning in adaptive control.
>
> Additionally, note that a controlled forgetting mechanism is important in online learning, because not all of the historical data might be useful (especially if the dynamics change over time) and it might be desirable to assign more weight to newer data. To this end, we used the bounded gain forgetting factor (i.e., $\beta$) mechanism from Slotine and Li's composite adaptive control method. Specifically, the PE condition under higher levels of excitation yields a larger forgetting factor, thus giving more weight to newer data.

---

> > ### Comment · Reviewer_N4X4 · 2024-11-29
> > **Thanks for this response**
> >
> > Thanks for the careful reply! I am now aware of the relevance of the proposed approach to deep learning. I have increased the score to 8.
> >
> > In addition, offline learning is not always a drawback, such as the continuing learning of animals. I am looking forward to physical verification!

---

> > > ### Author Response · Authors · 2024-11-29
> > >
> > > Thanks again for the thorough review, constructive feedback, and insightful discussion which has helped us improve the quality of the paper and deepen our understanding of this field. We completely agree that offline learning is not always a drawback.

---

### Official Review · Reviewer_bqXC · 2024-10-26

**Soundness:** 3
**Presentation:** 3
**Contribution:** 3
**Rating:** 6
**Confidence:** 2

**Summary:**

This paper provides the first result on simultaneous online system identification and trajectory tracking control of nonlinear systems using adaptive updates for all layers of the DNN. The Lyapunov-based stability analysis is provided, which guarantees that the tracking error, state-derivative estimation error, and DNN weight estimation errors are uniformly ultimately bounded.

**Strengths:**

This is first application of the Jacobian of the DNN to develop simultaneous online system identification and control. The theoretical content of the paper is good. The literature research is sufficient.

**Weaknesses:**

The practical application of this method requires high computing resources and is not suitable for personal computers. The presence of measurement noise does not seem to be considered in the two simulation tests, which is unreasonable. Control inputs of two simulations should also be presented. Moreover, the nonlinear dynamics of the selected simulation system is weak.

**Questions:**

1)	The existence of measurement noise should be considered, which is common in practical engineering; Please provide the control inputs of two simulation tests;
2)	I wonder if this method is effective for highly dynamic systems like quadcopters?
3)	Provide more experimental details, such as control inputs, weights update, and the selected control parameters.

---

> ### Author Response · Authors · 2024-11-23
>
> We thank the reviewer for their appreciation of the novelty, theoretical contents, and literature research. While we acknowledge that updating all layers of the DNN in real-time may require high computational resources, many real-world applications, especially those in industrial and robotics settings, have access to more powerful computational platforms, where such methods can be feasibly deployed.
>
> Based on the reviewer's suggestion, we performed all of the simulations again with measurement noise, and the results in the simulation section are now updated accordingly. We have also included the control input plots in both simulation examples according to the reviewer's request.
>
> Regarding the comment about the nonlinear dynamics in simulations being weak or the question whether this method is effective for highly dynamic systems like quadcopters, please note that the dynamics of the UUV in Section 4.2 (described in further detail in Appendix D.2) are similar to the quadcopter dynamics (see Eq. (1) in (O' Connell et al, 2022) for example). Such similarity is because both systems have an Euler-Lagrange dynamics resulting from the dynamics of rigid body translation and rotation. The UUV dynamics contains strong nonlinearities resulting from centripetal-Coriolis effects and hydrodynamic damping effects. So the developed method is indeed effective for highly dynamic systems.
>
> The additional simulation details on control inputs, weights update, and the selected control parameters are provided in Appendix D.
>
> References:
>
> O’Connell, M., Shi, G., Shi, X., Azizzadenesheli, K., Anandkumar, A., Yue, Y. and Chung, S.J., 2022. Neural-fly enables rapid learning for agile flight in strong winds. Science Robotics, 7(66), p.eabm6597.

---

### Official Review · Reviewer_XnMo · 2024-11-03

**Soundness:** 2
**Presentation:** 2
**Contribution:** 2
**Rating:** 5
**Confidence:** 4

**Summary:**

This paper addresses a methodology for simultaneously performing online system identification for the plant system and adaptation in the feedback control logic. Under some technical assumptions, stability conditions are presented, more specifically, the asymptotic stability of the equilibrium point of the entire feedback control system is ensured.

**Strengths:**

The simultaneous approach to online system identification and adaptation in the control logic, addressed in this paper, is well-motivated and justified with attractive numerical experiments.  In addition, as claimed by the authors, the convergence analysis for the identification error and control error is novel.

**Weaknesses:**

Assumptions 1 and 2 are mathematically severe.  While this paper claims its contribution lies in simultaneous system identification and control, Assumption 1 implies that performing system identification can achieve arbitrary control performance (arbitrary system dynamics is realized by u= g^+(r - f(x,\dot{x})). In this sense, the problem addressed in this paper is not essentially simultaneous.

Furthermore, there are too many technical assumptions on the modeling accuracy, meaning the existence of \bar{vareplison}, \bar{\theta}, etc.

**Questions:**

Could you relax Assumptions 1 and 2?  In particular, as commented in Weakness, Assumption 1 is mathematically (and practically) severe.  The authors comment that the developed methods can be extended to underactuated systems. However the details of the extension are scarcely explained, and no theoretical analysis is provided.  The reviewer believes that the extension to underactuated cases and its convergence analysis should be the main contribution of this paper.

---

> ### Author Response · Authors · 2024-11-23
>
> We thank the reviewer for their appreciation of the motivation of our paper and the numerical experiments. However, we respectfully disagree with the reviewer on the severity of the assumptions, which we justify as follows.
>
> **Regarding fully-actuated dynamics**
>
> Many practical nonlinear systems such as robot manipulators and UUVs used in the simulation section, Stewart platforms, hexapod robots meet this assumption. This is a broad variety of applications where the developed method can directly be applied and Assumption 1 holds mathematically and practically. The development is focused on fully-actuated systems to better focus on our unique specific contribution, i.e., the composite adaptive design for DNNs. The reason we do not consider the underactuated case is because, even for underactuated systems with perfect model knowledge, there is no universal nonlinear controller that can be directly applied to every underactuated system. Instead, the the constructive Lyapunov-based control design procedure is unique for each underactuated system. For example, the nonlinear control design for a nonholonomic mobile robot is completely different from the design for quadrotors. However, the steps required to extend the DNN-based control development from a fully-actuated system to an underactuated system mostly involves standard tools (e.g., backstepping) without any challenges specific to composite adaptive DNN development. The tools such as backstepping are well-known since 1980s, so including such an extension for specific cases would add limited theoretical value or novelty and confuse our main contribution.
>
> **UPDATE:** Based on the reviewer's suggestion, we have revised the manuscript with a detailed illustration on extending the development for a nonholonomic mobile robot in Appendix F. This extension does not require taking the pseudo-inverse according to Assumption 1 but instead achieves tracking by backstepping the kinematics into dynamics based on the development in (Fierro, Lewis, 1998) work. We hope this addresses the reviewer's concerns about Assumption 1.
>
> **Regarding simultaneity of system identification and control**
>
> It appears there is possibly a misunderstanding regarding what we mean by simultaneous system identification and control. The term $f(x,\dot{x})$ is unknown and not being used anywhere in the controller. The control input is not designed as $u=g^{+}(r-f(x,\dot{x}))$, but $u=g^{+}(x,\dot{x})(\\ddot{x}\_\{d}(t)-(\alpha_\{1}+k_\{r})r+(\alpha_\{1}^{2}-1)e-\Phi(X,\hat{\theta}))$ according to Eq. (6). The term $\Phi(X,\hat{\theta})$ is a DNN-based adaptive estimate of $f(x,\dot{x})$, updated in real-time. Specifically, the DNN parameter estimate $\hat{\theta}$ updates in real-time based on the adaptive update law $\dot{\hat{\theta}}=\mathrm{proj}(-k_{\hat{\theta}}\Gamma(t)\hat{\theta}+\Gamma(t)\Phi^{\prime\top}(X,\hat{\theta})(r+\alpha_{3}E))$. Therefore, the control and adaptive update laws operate simultaneously, unlike offline training methods which identify the parameter estimates a priori using training datasets. Assumption 1 or fully-actuatedness does not violate this simultaneity.
>
> **Regarding Assumption 2**
>
> The assumption is not about the existence of $\bar{\theta}$ but about its knowledge. Note that $\bar{\theta}$ is a bound on the unknown constant ideal parameter $\theta^*$, so $\bar{\theta}$ exists because $\theta^*$ is a constant. This is a standard mathematical assumption in NN-based adaptive control literature (e.g., see (Lewis, 1996)). As far as the approximation accuracy $\bar{\varepsilon}$ is concerned, its existence is not an assumption but a fact because due to the universal function approximation property of DNNs. In fact, even without the universal function approximation property, a $\overline{\varepsilon}>0$ satisfying $\sup_{X\in\Omega}\left\Vert f(x,\dot{x})-\Phi(X,\theta^{*})\right\Vert \leq\overline{\varepsilon}$ would exist because the function $f$ and $\Phi$ are continuous and therefore bounded over the compact set $\Omega$. The universal function approximation property further allows $\overline{\varepsilon}$ to be prescribed as arbitrarily small.
>
> Based on the reviewer's comment, we revised Section 2.1 to state, “Assumption 2 is reasonable since in practice the user can select $\overline{\theta}$ a priori to restrict the parameter search space. If such a selection does not obey Assumption 2, the selection may no longer allow the user to make $\bar{\varepsilon}$ arbitrarily small as guaranteed by the universal function approximation property. However, a bound $\bar{\varepsilon}$ satisfying $\sup_{X\in\Omega}\left\Vert \varepsilon(X)\right\Vert \leq\overline{\varepsilon}$ still exists due to the continuity of $f$ and $\Phi$ over $\Omega$. Using heuristic approaches, if such $\bar{\varepsilon}$ is found to be larger than the maximum allowable error, then $\bar{\theta}$ can be iteratively increased until it achieves the prescribed $\bar{\varepsilon}$.”

---

> > ### Author Response · Authors · 2024-11-23
> >
> > References:
> >
> > F. L. Lewis, A. Yegildirek, and Kai Liu. Multilayer neural-net robot controller with guaranteed tracking performance. IEEE Trans. Neural Netw., 7(2):388–399, March 1996a. 10.1109/ 72.485674.
> >
> > R. Fierro and Frank L. Lewis. Control of a nonholonomic mobile robot using neural networks. IEEE Trans. Neural Netw., 9(4):589–600, July 1998
> >
> > Shi, G., Shi, X., O’Connell, M., Yu, R., Azizzadenesheli, K., Anandkumar, A., Yue, Y. and Chung, S.J., 2019, May. Neural lander: Stable drone landing control using learned dynamics. In 2019 international conference on robotics and automation (icra) (pp. 9784-9790). IEEE.

---

> > ### Comment · Reviewer_XnMo · 2024-11-26
> > **Thanks for revision and response**
> >
> > The reviewer agrees the response "The reason we do not consider the underactuated case is because, even for underactuated systems with perfect model knowledge, there is no universal nonlinear controller that can be directly applied to every underactuated system. Instead, the the constructive Lyapunov-based control design procedure is unique for each underactuated system."
> >
> > The reviewer correctly understands that the control input is not designed as $u=g^{+}( X -f(x,\dot{x}))$ by using some X of generating arbitrary dynamics.  He/she states that Assumption 1 severely limits the class of the plant system to completely eliminate both the challenge (and value) of the simultaneous problem.
> >
> > The reviewer was satisfied with the revision regarding Assumption 2. Thank you for the update.

---

> ### Author Response · Authors · 2024-11-25
> **Update: Underactuated Nonholonomic Mobile Robot System Example Added in Appendix F**
>
> Based on the reviewer's suggestion to provide the control development for under-actuated cases, we have revised the manuscript with a detailed illustration on extending the development for a nonholonomic mobile robot in Appendix F. This extension does not require taking the pseudo-inverse according to Assumption 1 but instead achieves tracking by backstepping the kinematics into dynamics based on the development in (Fierro, Lewis, 1998) work. We hope this addresses the reviewer's concerns about Assumption 1.

---

> ### Author Response · Authors · 2024-11-26
>
> We thank the reviewer for the timely response. We are happy the reviewer found our revisions regarding Assumption 2 satisfactory.
>
> As for Assumption 1, it is not clear if the reviewer noticed our more recent update in Appendix F, where we added a brief illustration on extending the procedure to underactuated nonholonomic mobile robots. This example was added based on the reviewer's suggestion to provide more insights on performing the extension to an underactuated case.
>
> Since the reviewer agrees with the response "the constructive Lyapunov-based control design procedure is unique for each underactuated system", we hope the unrealisticness of the expectation to consider a general underactuated system is understood. The extension needs to be performed on a case-by-case basis. However, the process of performing such an extension mostly involves standard nonlinear control tools that are not unique to our work. Specifically, note that the composite adaptive DNN (our main contribution) part of the development in Appendix F is mostly the same as in Section 2 of the manuscript. The part of the development specific to mobile robots involved standard nonlinear control insights (i.e.,backsteppping approach well known in literature for mobile robots) from (Fierro, Lewis, 1998) work. The extension to an underactuated case did not involve any theoretical novelty. To better focus on our main contribution, we focus the main development on non-underactuated systems.
>
> Additionally, we still think there is a possible misunderstanding regarding simultaneity. The reviewer is requested to clarify on the statement in the original review,
>
> > "Assumption 1 implies that performing system identification can achieve arbitrary control performance (arbitrary system dynamics is realized by $u= g^+(r - f(x,\dot{x}))$. In this sense, the problem addressed in this paper is not essentially simultaneous."
>
> The briefness of the reviewer's comments is making it difficult for us to interpret what was being meant. Assumption 1 is not precluding the simultaneity of identifying $f(x,\dot{x})$ and implementing the control law. The control and adaptive update laws are simultaneous. The reviewer is requested to clarify if we are on the same page with that.
>
> >"Assumption 1 severely limits the class of the plant system to completely eliminate both the challenge (and value) of the simultaneous problem"
>
> We would like to bring the reviewer's attention to the challenges involved in the DNN part of the development, which is the focus of our main contribution. There were multiple challenges precluding this development, all of which are explained in great detail in the Introduction and Appendix B. Specifically, developing Lyapunov-based composite adaptive update laws for all
> DNN layers is a challenging problem especially due to the nested and nonlinear-in-parameter structure. Specifically, the term $\Phi^{\prime}\left(X,\hat{\theta}\right)$ is not the standard regressor in adaptive control, but the DNN's Jacobian with respect to its vectorized weights, an analytical derivation of which is provided in Appendix C. The DNN's Jacobian is derived for an arbitrary number of layers by leveraging its recursive compositional structure. Additionally, the formulation of the prediction error term E has never been done in literature for a DNN because of technical challenges that are specific to a DNN. Slotine's result ([24] in the previous version) used a filtered regression approach to formulate the prediction error. As described in Appendix B.2, there are mathematical challenges in extending Slotine's filtered regression approach to DNNs because the approach utilized the LIP structure of the model. To circumvent those challenges, we used a dynamic state-derivative estimator to construct a new prediction error, and the corresponding estimation errors are also incorporated in the Lyapunov-based analysis to ensure closed-loop stability after combining the developed controller, adaptation laws, and state-derivative estimator.
>
> Therefore, even with Assumption 1, the challenges are still significant, which the other reviewers have recognized. Note that this is a very recent topic of research with only a few closely related works, e.g., (Patil et. al., 2022) and (O' Connell et. al., 2021), both highly cited works. Both of these works performed the development assuming fully-actuated dynamics. If Assumption 1 completely eliminated the challenges like the reviewer claims, this topic of research would not have emerged in the first place.
>
> Patil, O.S., Le, D.M., Greene, M.L. and Dixon, W.E., 2021. Lyapunov-derived control and adaptive update laws for inner and outer layer weights of a deep neural network. IEEE Control Systems Letters, 6, pp.1855-1860.
>
> O’Connell, M., Shi, G., Shi, X., Azizzadenesheli, K., Anandkumar, A., Yue, Y. and Chung, S.J., 2022. Neural-fly enables rapid learning for agile flight in strong winds. Science Robotics, 7(66), p.eabm6597.

---

### Official Review · Reviewer_vqH4 · 2024-11-03

**Soundness:** 3
**Presentation:** 3
**Contribution:** 3
**Rating:** 8
**Confidence:** 3

**Summary:**

This paper is introducing an adaptive DNN-based controller. Standard adaptive DNN-based controllers are based on Lyapunov-based analysis and allow updates on the last layers of the NNs only. They only use the tracking error as a metric to indicate when adaptation is needed, and only provide guarantees on the tracking error convergence. This paper introduces a dual (composite) method, for continuous system identification combined with trajectory tracking, and guarantees that the tracking error, state-derivative estimation error, and DNN weight estimation errors are uniformly ultimately bounded. The last two reflect identifying the dynamics of the system.
The system identification is performed via a dynamic state-derivative estimator and under the assumption of persitence of excitatiton. The controller is evaluated in simulation on a two-link manipulator system and an unmanned underwater vehicle system with intermittent loss of state feedback, and shows improvement compared to baseline methods.

**Strengths:**

The paper is on a very timely topic, DNN-based control. With the emergence of AI-based approaches, a principled treatment of a learning-based controller is an impactful contribution. Introducing a controller which allows for updates in all layers, in contrast to the state of the art where only thelast layers can be updated, and where system dynamics is not considered explicitely, is a big contribution.
Given the complexity of the paper, it is very clearly presented. The simulation examples are relevant and not just datasets, but dynamical systems.

**Weaknesses:**

1. In the simulations, there is only comparison to DNN-based controllers.
It would highly interesting to see how is the peroformance, compared to more standard controllers used in robotics (for example, MPC-based, or RL-based controllers). I am willing to raise my rating if this is added.

**Questions:**

1. Shouldn’t there be an assumption for the activation functions to be convex? How do you deal with the nonconvex dependence of the underlying loss function on weights of hidden layers? Is it playing a role?

2. How can input and state constraints be integrated in the proposed approach?

---

> ### Author Response · Authors · 2024-11-23
>
> We thank the reviewer for their appreciation of the strength, timeliness, and potential impact of our contribution. Based on the reviewer's suggestion to also include more standard controllers used in robotics as baselines, we have updated the simulation section to include nonlinear MPC and nonlinear PD controllers as baselines for both examples, the robot manipulator and the UUV. The developed method was able to achieve improved tracking performance with a similar control effort compared to both of these baselines. Furthermore, while we acknowledge that the comparison with nonlinear MPC provides valuable context, it is important to note that our approach operates under a fundamentally different paradigm. Specifically, using DNN-based online system identification, our method addresses scenarios where prior model knowledge is unavailable or unreliable, highlighting its practical applicability in uncertain environments. We hope this clarifies the distinction and demonstrates the merits of our approach. In future work, we also plan to extend our approach to develop a DNN-based adaptive MPC method, where the DNN-based adaptive system identifier can update the model in real-time for improving future state predictions over the horizon.
>
> **Regarding nonconvex dependence**
>
> The convexity condition on an underlying loss did not appear in our analysis because we used a Lyapunov-based approach instead of traditional optimization-based approaches involving a loss function. Instead, we require the persistence of excitation (PE) condition and some gain conditions that may be considered analogous to a convexity condition. Specifically, instead of imposing convexity on the DNN, we use a first-order Taylor series approximation $\Phi(X,\theta^{*})-\Phi(X,\hat{\theta})=\Phi^{\prime}(X,\hat{\theta})\tilde{\theta}+\mathcal{O}(\left\Vert \tilde{\theta}\right\Vert ^{2})$ as in Eq. (14) of the paper, where the higher-order term $\mathcal{O}(\left\Vert \tilde{\theta}\right\Vert ^{2})$ is shown to be bounded when $z\in\mathcal{D}$ and because $\left\Vert \tilde{\theta}\right\Vert \leq2\bar{\theta}$ due to the projection operator. The bound on $\mathcal{O}(\left\Vert \tilde{\theta}\right\Vert ^{2})$ carries forward to the gain conditions. Recall that $$\lambda_3=\min \\{ \alpha_{1},k_{r}-\frac{\gamma_{1}}{2},\alpha_{2},k_{f}-\frac{\gamma_{2}+\alpha_{3}\gamma_{3}}{2},\frac{k_{\hat{\theta}}}{2}+\frac{\beta_{1}}{2\varkappa_{0}}-\alpha_{3}\gamma_{3} \\}$$.
> As mentioned in Remark 3, the gains $\alpha_{1}$,$\alpha_{2}$,$\alpha_{3}$,$k_{r}$, and $k_{f}$ can be selected to be sufficiently high such that $\lambda_{3}=\frac{k_{\hat{\theta}}}{2}+\frac{\beta_{1}}{2\varkappa_{0}}-\alpha_{3}\gamma_{3}$; thus, $ \frac{\lambda_{2}c}{\lambda_{1}\lambda_{3}}=\frac{\lambda_{2}}{\lambda_{1}}\left(\frac{\gamma_{1}+\gamma_{2}+k_{\hat{\theta}}\bar{\theta}^{2}+\alpha_{3}\gamma_{3}\gamma_{1}^{2}}{k_{\hat{\theta}}+\frac{\beta_{1}}{\varkappa_{0}}-2\alpha_{3}\gamma_{3}}\right)$. Since the term $\beta_{1}$ is quantitatively indicative of the PE condition with larger values under more excitation, the bound $\sqrt{\frac{\lambda_{2}c}{\lambda_{1}\lambda_{3}}}$ becomes tighter under more excitation and can be arbitrarily decreased under higher gains and more excitation. Under the absence of PE, a bound is still obtained by selecting an appropriate high $k_{\hat{\theta}}>2\alpha_{3}\gamma_{3}$.
>
> **Regarding Input and State Constraints**
>
> The input and state constraints can be incorporated using control barrier functions (CBFs) (Ames et. al., 2016). Specifically, the control input from the developed method can be passed as a nominal control input through a CBF-based safety filter involving a quadratic program with CBF-based constraints. However, exploring input and state constraints is beyond the scope of this work as the CBF-based safety constraints are either difficult to obtain or too conservative under the lack of prior model knowledge. We plan to investigate DNN-based adaptive CBFs in future work, where safety constraints can be gradually made less conservative as the DNN learns the system dynamics online. Based on the reviewer's comment, we have added the following sentence in Section 6 to discuss future work:
>
> "Furthermore, extensions of the developed online system identification approach in optimization-based control paradigms such as MPC and reinforcement learning can be explored. Moreover, future research efforts can also investigate how to combine the developed method with control barrier functions to satisfy state and input constraints."
>
> **References:**
>
> Ames, A.D., Xu, X., Grizzle, J.W. and Tabuada, P., 2016. Control barrier function based quadratic programs for safety critical systems. IEEE Transactions on Automatic Control, 62(8), pp.3861-3876.

---

> > ### Comment · Reviewer_vqH4 · 2024-11-25
> >
> > Thank you for providing the simulations comparing to nonlinear MPC and nonlinear PD. Indeed, DNN-based methods would address the situation with limited model knowledge; however, standard control methods could also be adapted to such a situation (MPC+mismatch learning, etc.).
> > The clarification about the nonconvex dependence is useful.
> > I am increasing my rating.

---

> > > ### Author Response · Authors · 2024-11-25
> > >
> > > Thanks again for the helpful review which has significantly improved the quality of our paper! Indeed, MPC with mismatch learning can be adapted to such a situation. We will keep this in mind while investigating future extensions of our work to the MPC paradigm. Thanks for this information!

---

### Author Response · Authors · 2024-11-24
**Summary of Changes**

We are grateful to all the reviewers for their positive and constructive feedback. We appreciate that all reviewers acknowledge the contribution of our work: **"the paper is on a very timely topic", "a principled treatment of a learning-based controller is an impactful contribution", "introducing a controller which allows for updates in all layers, in contrast to the state of the art where only the last layers can be updated, and where system dynamics is not considered explicitly, is a big contribution"** (reviewer vqH4), **"... is well-motivated and justified with attractive numerical experiments"** (reviewer XnMo), **"This is first application of the Jacobian of the DNN to develop simultaneous online system identification and control. The theoretical content of the paper is good. The literature research is sufficient."** (reviewer bqXC) , **" some novel theoretical results are developed in this paper with rigid proofs. The presentation is also clear"** (reviewer N4X4).

All changes in the paper have been marked in blue. For the convenience of the reviewers, we are providing a summary of revisions in the manuscript below.

**1. More baselines:** The robot manipulator and UUV simulations (Sections 4.1 and 4.2, respectively) are now expanded with nonlinear MPC, nonlinear PD, and observer-based disturbance rejection controllers as additional baselines for simulations.

**2. Measurement Noise:** All of the simulations were performed again incorporating state measurement noise to make the simulations more realistic. Sections 4.1 and 4.2 have been updated accordingly.

**3. More Details on Simulations:** Based on the reviewers' requests, additional simulation details on control inputs, weights update, and the selected control parameters along with their selection strategy are provided in Appendix D.

**4. Expanded Justification of Assumptions:** More discussion is added in Section 2 to justify Assumption 1 and 2.

**5. Further Discussion on Related Works:** A more detailed discussion regarding the related work (O'Connell, 2022) is added in Appendix B.2.

**6. Modified Abstract:** The abstract is made more concise and the factors that determine the upper bounds of convergence sets are provided.

**7. Future Work:** More discussion is added in Section 6 on possible future extensions of this work to incorporate state and input constraints.

**8. Extension to underactuated case of nonholonomic mobile robot:** An extension of the development to nonholonomic mobile robots is added in Appendix F.

---
**Summary of Discussion Period**

1. **Reviewer vqH4** suggested incorporating comparisons with more standard baselines, such as Model Predictive Control (MPC), and asked questions about convexity. In response, we **added comparisons with nonlinear MPC and nonlinear PD as baselines** and provided detailed explanations of how conditions analogous to convexity emerge in the form of Persistent Excitation (PE) and gain conditions in the Lyapunov analysis. The reviewer was satisfied with these additions and **raised the score to 8**.

2. **Reviewer XnMo** critiqued Assumptions 1 and 2, requesting more justification and **details on extending the method to underactuated systems** along with a theoretical analysis. We addressed these concerns by providing stronger justifications for the assumptions and clarifying that **theoretical extensions of any controller to underactuated systems are not universal but inherently case-specific**. To illustrate the process of such extension, we **included an extension to a nonholonomic mobile robot** in Appendix F. While the reviewer was satisfied with our justification for Assumption 2, they have not yet commented on the extension for mobile robots. Furthermore, the reviewer stated Assumption 1 eliminates the challenges, which we disagree with. **The challenges we addressed are in the DNN all-layer update part, our key contribution**, which the reviewer did not comment on yet.

3. **Reviewer bqXC** recommended incorporating measurement noise in the simulations and adding more experimental details. **We revised the manuscript accordingly.** Additionally, the reviewer inquired whether the method could handle strongly nonlinear systems like quadcopters. We affirmed that it could, noting that the Unmanned Underwater Vehicle example in our work includes nonlinear dynamics comparable to those of a quadcopter. The reviewer acknowledged our rebuttal, remarked that 6 is an appropriate rating for our current manuscript, and gave constructive feedback on improving the quality of our work in future.

4. **Reviewer N4X4** suggested improvements to the related work section, simulations, and abstract. We implemented most of their recommendations in our revisions. The reviewer raised follow-up questions about the **connections between deep learning and adaptive control** in the context of our contribution. Our detailed responses convinced the reviewer, leading to an **increased score of 8**.

---

### Meta-Review · Area_Chair_uSRK · 2024-12-16

**Metareview:**

This paper demonstrates a result for simultaneous identification and stabilization of a control system with a deep neural network. The reviewers were generally positive, but with one reviewer leaning towards rejection. The positive reviewers were rather lacking in detail, whereas the negative reviewer demonstrated greater expertise. In addition, reviewers did not actively participate in discussion.

Therefore, I decided to intervene as meta-reviewer to assess the paper on its own merits.
(a) this paper establishes a simultaneous tracking and stabilization for nonlinear systems. They claim this is novel, because classical results only apply to linear parameter uncertainty.
(b) This paper mostly makes good on its claim
(c) I believe the presentation of the paper is below the standards of ICLR, and will be somewhat impenetrable for the community. For one, results rely on a number of terms which are defined haphazardly throughout. Second, certain constants (e.g. the Taylor expansion should depend on the smoothness of G) are omitted. Moreover, NNs are typically highly non-smooth, so framing arguments as Taylor expansions seem misleading. Third, the authors claimed limitation of Slotine and Li, '89 is that they require a low pass filter, which necessitates LPV systems. But I do not seem to see any form of filtering in the paper, and indeed the authors seem to act as if they have access to time derivatives. Moving beyond this point is quite poorly explained.
(d) Ultimately, it seems that the theoretical of this content has little to do with neural networks at all, and should be about tracking/estimation with non-LPV uncertainty. The presentation has great room for improvement, and the reviewer who appeared most knowledgable also seems to side with rejection.

Constructive suggestions to the authors: (a) rewrite proofs, exposition, clean up theorem statements and remove excess and unnecessary terms. (b) I would suggest submitting to a revised manuscript to a controls venue, where the work can be appropriately evaluated by a community with a more extensive knowledge of nonlinear adaptive control.

**Additional Comments On Reviewer Discussion:**

The negative reviewer concern still remained.

---

### Decision · Program_Chairs · 2025-01-22

Reject